



# Estimates of critical loads and exceedances of acidity and
# nutrient nitrogen for mineral soils in Canada for 2014–2016
# average annual sulphur and nitrogen atmospheric deposition
Hazel Cathcart[1], Julian Aherne[2], Michael D. Moran[1*], Verica Savic-Jovcic[1], Paul A. Makar[1], and
Amanda Cole[1].
[1]Air Quality Research Division, Atmospheric Science and Technology Directorate, Science and Technology Branch,
Environment and Climate Change Canada, Toronto, Ontario, M3H 5T4, Canada
[2]School of the Environment, Trent University, Peterborough, Ontario, K9L 0G2, Canada
*Scientist emeritus
*Correspondence to*: Hazel Cathcart (hazel.cathcart@ec.gc.ca)
**Abstract.** The steady-state Simple Mass Balance model was applied to natural and semi-natural terrestrial
ecosystems across Canada to produce nation-wide critical loads of acidity (maximum sulphur, $CL_{max}S$; maximum
nitrogen, $CL_{max}N$; minimum nitrogen, $CL_{min}N$) and nutrient nitrogen ($CL_{nut}N$) at 250 m resolution. Parameterization
of the model for Canadian ecosystems was considered with attention to the selection of the chemical criterion for
damage at a site-specific resolution, with comparison between protection levels of 5% and 20% growth reduction
(approximating commonly chosen base-cation-to-aluminum ratios of 1 and 10 respectively). Other parameters
explored include modelled base cation deposition and site-specific nutrient and base cation uptake estimates based
on North American tree chemistry data and tree species and biomass maps. Soil critical loads of nutrient nitrogen
were also mapped using the Simple Mass Balance model. Critical loads of acidity were estimated to be low (e.g.,
below 500 eq$^{-1}$ ha yr$^{-1}$) for much of the country, particularly above 60°N latitude where base cation weathering rates
are low due to cold annual average temperature. Exceedances were mapped relative to annual sulphur and nitrogen
deposition averaged over 2014–2016. Results show that under a conservative estimate (5% protection level), 10%
of Canada's Protected and Conserved Areas in the study area experienced exceedance of some level of soil critical
load of acidity while 70% experienced exceedance of soil critical load of nutrient nitrogen.
**1 Introduction**
During the last three decades, reductions in sulphur (S) and nitrogen (N) emissions and acidic deposition have led to
improvements in ecosystem health across the U.S. and Canada; nonetheless, the acid rain question remains relevant
in Canada. Large point sources of emissions in western Canada have emerged, prompting concerns of impacts to
sensitive ecosystems in British Columbia and the Athabasca Oil Sands Region (AOSR) in northeastern Alberta (e.g.,
Mongeon et al., 2010; Williston et al., 2016; Makar et al., 2018). Further, increased marine traffic in the Arctic due
to the effects of anthropogenic warming has raised questions about potential impacts of acidic deposition on
northern ecosystems already under pressure from climate change (Forsius et al., 2010; Liang and Aherne, 2019).
Recovery of forest soils from decades of elevated acidic deposition in the northeastern U.S. and eastern Canada is



encouraging, but is predicted to be slow (Lawrence et al., 2015; Hazlett et al., 2020) and is complicated by the effect
of elevated N deposition (Clark et al., 2013; Simkin et al., 2016; Pardo et al., 2019; Wilkins et al., 2023) and climate
change (Wu and Driscoll, 2010).  The importance of N deposition to acidification and eutrophication has received
increased recognition in recent years, prompting new avenues of risk assessment and mapping (e.g. empirical critical
loads of nitrogen; (Bobbink et al., 2022; Bobbink and Hicks, 2014).  While N oxide emissions in Canada declined
by 41% between 1990 and 2022, ammonia emissions increased by 24% in that same period (ECCC, 2024).

The critical loads concept, defined as "the maximum deposition that will not cause chemical changes leading to
long-term harmful effects on ecosystem structure and function" (Nilsson and Grenfelt, 1988) is the primary tool for
identifying ecosystems that are sensitive to air pollution, particularly with respect to acidification and eutrophication
(De Vries et al., 2015; Burns et al., 2008).  Ecosystems that receive acidic deposition above their critical load are
said to be in exceedance; that is, they are at risk of undergoing biological damage.  Soil acidification is characterized
by attrition of base cations and a decrease in soil pH, which in turn causes leaching of toxic metals, such as
aluminum, and damage to plant roots.  During the past three decades, these effects have been observed in forest soils
in the northeastern U.S. and eastern Canada (e.g., Cronan and Schofield, 1990; Likens et al., 1996; Lawrence et al.,
1999) that received acidic deposition in excess of their critical loads.  The effects of nutrient N on ecosystems,
which include eutrophication, reduced plant biodiversity, and plant community changes, have also become an
emerging issue, with studies suggesting that some Canadian ecosystems are in exceedance of their nutrient N critical
load (e.g., Aherne and Posch, 2013; Reinds et al., 2015; Williston et al., 2016).

The standard approach for estimating soil critical loads is the Simple Mass Balance (SMB) model (Sverdrup and De
Vries, 1994, Posch et al., 2015), a steady-state model with several simplifying assumptions to reduce input
requirements.  This approach has been used for regional and provincial critical load assessments in Canada (e.g.,
Ouimet et al., 2006; Aklilu et al., 2022) as well as on a multi-provincial (NEG-ECP, 2001; Carou et al., 2008;
Aherne and Posch, 2013) and national scale (Reinds et al., 2015).  However, nationwide implementations of the
SMB model in Canada have been challenged by data paucity and disharmony across provinces (i.e. different data
sources, methodology and spatial alignment), coarse input map resolution, and computational difficulties driven by
the size of the country and the subsequent size of data files used in critical load calculations.  In recent years,
though, high-resolution input data (for soils, meteorology, and forest composition) have become available and
present an opportunity to refine, expand, and harmonise critical loads across the entire country, including extending
maps into the Canadian Arctic.  These developments come at a time when policymakers in Canada are seeking to
define and track air quality impacts (such as those by acidic S and N deposition) on sensitive ecosystems under the
Addressing Air Pollution Horizontal Initiative (ECCC, 2021).  Furthermore, development of high-resolution critical
loads of nutrient N to assess terrestrial eutrophication risk may contribute to efforts to meet biodiversity goals such
as those under the Kunming-Montreal Global Biodiversity Framework (ECCC, 2023c).  While the SMB model is a
well-established and widely used approach to determine critical loads, there remains a need for harmonised



application across Canadian ecosystems to provide maps from which the effect of S and N deposition can be estimated.

The objective of this study was to assess the impacts of acidic and nutrient N deposition on terrestrial ecosystems Canada-wide using the critical loads framework. In doing so, we applied a harmonised methodology to the SMB model for Canadian ecosystems using high-resolution input maps, including modelled Canada-wide base cation deposition (crucial for the estimation of critical loads). We also explored the choice of chemical (damage) criterion for Canadian ecosystems using a site-specific approach. Finally, we assessed the impact of anthropogenic base cation deposition on exceedance estimates under annual average S and N deposition (ECCC, 2023a) for the three-year period 2014–2016, using the Canadian Protected and Conserved Areas Database (CPCAD; ECCC, 2023b), to evaluate risk to sites that may be of interest to policymakers.

## 2 Methods

### 2.1 Study area

As the second-largest country by landmass in the world at over 9.9 million km$^2$, Canada is home to a variety of climates, soils, vegetation, and geological structures that are often grouped into distinct ecozones (Figure 1A). The full extent of Canada was included in this study to bring together estimates for all 10 provinces and 3 territories. However, only natural and semi-natural soils meeting certain criteria for critical load estimations were considered. A land cover map (CEC, 2018) was used to exclude non-soil ecosystems including water, wetlands, and permanent snow and ice (see Figure 1B). Soils were further limited to natural and semi-natural ecosystems by excluding urban areas, crop classes, and areas within the boundaries of the agricultural ecumene (Figure 1B). Areas considered "barren" by land classification were not excluded when soil depth was indicated. Since peat and wetland soil classification is difficult at a Canada-wide scale (i.e. data at the required scale are presently unavailable), organic soils with 30% or more organic matter content were filtered out. The Hudson Plain ecozone (which contains the world's largest contiguous wetland) was also broadly excluded from the study because of very low mineral soil coverage.

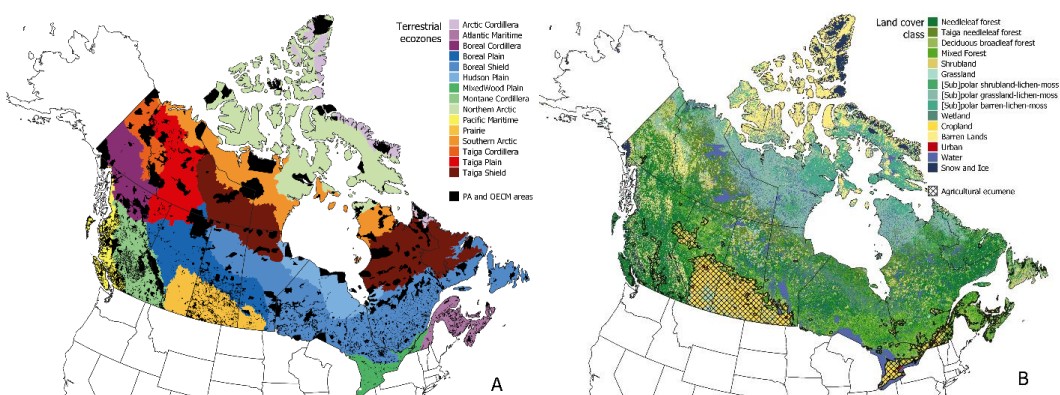

**Figure 1: Study area illustrating the 15 terrestrial ecozones of Canada (A, data source: Agriculture and Agri-Food**
**Canada, 2013) with terrestrial Protected and Other Effective area-based Conservation Measures (OECM) areas in black**
**(ECCC, 2023b), as well as 15 land cover classes (B, data source: CEC, 2018) with agricultural regions in crosshatch**
**(Statistics Canada, 2017).**

**2.2 The Simple Mass Balance model for acidity and nutrient nitrogen**
Critical loads of acidity were estimated using the SMB model, which balances sources, sinks, and outflows of S and
N in terrestrial ecosystems while assuming ecosystems are at long-term equilibrium (i.e. about 100 years,
representing at least one forest rotation cycle) (CLRTAP, 2015). The SMB model defines the critical load of S and
N acidity (Figure 2) as a function of the maximum S critical load ($CL_{max}S$), the maximum N critical load ($CL_{max}N$),
and the amount of N taken up by the ecosystem ($CL_{min}N$). Pairs of S and N deposition that fall outside this function
(white region, Figure 2) signify that the receiving ecosystem is in exceedance of its critical load of acidity (i.e., it
receives a potentially damaging amount of acidic deposition).

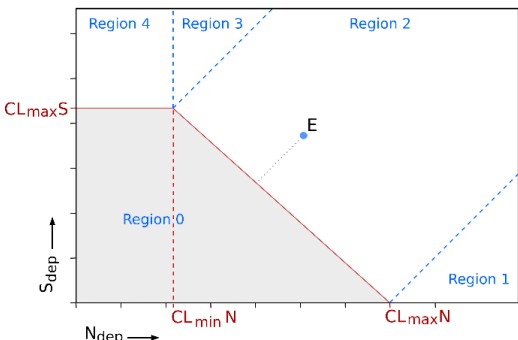


**Figure 2: The acidity critical load function (red line) is defined by the maximum sulphur critical load ($CL_{max}S$), the**
**maximum nitrogen critical load ($CL_{max}N$) and the minimum nitrogen critical load ($CL_{min}N$). Deposition points falling**
**outside the critical load function (e.g., point E) are in exceedance (and defined as Regions 1-4), while those within the grey**
**area (Region 0) are protected.**



The determination of $CL_{max}S$ requires knowledge of non-sea salt base cation (calcium, magnesium, potassium,
sodium) deposition ($BC_{dep}$), soil base cation weathering ($BC_{we}$), chloride deposition ($Cl_{dep}$), base cation uptake
($Bc_{up}$), and the critical leaching of Acid Neutralizing Capacity (the ability of the ecosystem to buffer incoming
acidity), denoted $ANC_{le,crit}$ (see Eq. 1).  Note that sodium is included in some base cation terms (denoted BC, e.g.,
$BC_{we}$) when sodium contributes to buffering, but where it concerns uptake by vegetation sodium is omitted since it
is non-essential to plants (denoted Bc, e.g., $Bc_{up}$).

$$CL_{max}S = BC_{dep} + BC_{we} - Cl_{dep} - Bc_{up} - ANC_{le,crit}$$   (1)

The value of $ANC_{le,crit}$ (see Eq. (2)) is determined from a critical base-cation-to-aluminum ratio ($Bc/Al_{crit}$), which is
set to protect the chosen biota within ecosystems of interest (i.e., the critical chemical criterion), soil percolation or
runoff (Q), and the gibbsite equilibrium constant ($K_{gibb}$).

$$ANC_{le,crit} = -Q^{\frac{2}{3}} \cdot \left( 1.5 \cdot \frac{Bc_{dep} + Bc_{we} - Bc_{up}}{K_{gibb} \cdot \left(\frac{Bc}{Al}\right)_{crit}} \right)^{\frac{1}{3}} - \left( 1.5 \cdot \frac{Bc_{dep} + Bc_{we} - Bc_{up}}{\left(\frac{Bc}{Al}\right)_{crit}} \right)$$   (2)

The calculation of $CL_{min}N$ from Eq. (3) describes the limit above which N deposition becomes acidifying, where $N_u$
denotes N taken up by vegetation and $N_i$ denotes long-term net immobilization of N in the root zone of soils under
steady state conditions.  A value of 35.714 eq ha$^{-1}$ yr$^{-1}$ (0.5 kg N ha$^{-1}$ yr$^{-1}$) was used, based on estimates of annual $N_i$
since the last glaciation by Rosen et al. (1992).  Lastly, $CL_{max}N$ is estimated from Eq. (4) using $CL_{max}S$, $CL_{min}N$, and
the soil denitrification (the loss of nitrate to nitrogen gas) factor ($f_{de}$).

$$CL_{min}N = N_i + N_u \qquad ,$$   (3)

$$CL_{max}N = CL_{min}N + \left( \frac{CL_{max}S}{1 - f_{de}} \right) \qquad .$$   (4)

Equation (5) was used to estimate soil critical loads of nutrient N ($CL_{nut}N$), wherein the acceptable inorganic N
leaching limit, a value set to prevent harmful effects of nutrient N such as eutrophication, vegetation community
changes, nutrient imbalances, and plant sensitivity to stressors, is set from acceptable N concentrations in soil
solution ($[N]_{acc}$) multiplied by Q (CLRTAP 2015).  The $[N]_{acc}$ was set to 0.0142 eq m$^{-3}$ (0.2 mg N l$^{-1}$ in soil solution)
for conifer forests and 0.0214 eq m$^{-3}$ (0.3 mg N l$^{-1}$) for all other semi-natural vegetation, following the generalised
approach taken for the European critical loads database (Reinds et al., 2021) as values suggested in CLRTAP (2015)
are often country-specific and do not extend to other regions or ecosystems.

$$CL_{nut}N = N_i + N_{up} + \left( \frac{Q * [N]_{acc}}{1 - f_{de}} \right) \qquad .$$   (5)



**2.3 Data and mapping**

Critical load estimates were calculated with the statistical programming language R, wherein inputs (Table 1) to and outputs from the SMB model were represented by 250 m resolution raster maps. Alignment and projection in WGS84 followed the layers sourced from the OpenLandMap.org project (i.e., Hengl (2018c, a, d, b); Hengl and Wheeler (2018) in Table 1), since they represented the majority of input (raster) data sources. Output maps were visualised using QGIS (QGIS Development Team, 2023) with accessible colour schemes (Tol, 2012). Acidity critical load components ($CL_{max}S$, $CL_{max}N$, $CL_{min}N$) and $CL_{nut}N$ were all mapped using equivalents of acidity (or nutrient nitrogen) per hectare per year (eq ha$^{-1}$ yr$^{-1}$).

**Table 1: Data sources for input parameters to the SMB model and critical load exceedance calculation.**

| Parameter | Units | Use | Original resolution | Source |
|---|---|---|---|---|
| Temperature | | | | |
| Average annual air temperature (1981–2010) | °C | $BC_{we}$ | 250 m | McKenney et al., 2006 |
| Soil | | | | |
| Absolute depth to bedrock | cm | $BC_{we}$ | 250 m | Hengl, 2017 |
| Organic carbon | × 5 g kg$^{-1}$ | $BC_{we}$ | 250 m | Hengl & Wheeler, 2018 |
| Sand fraction | % | $BC_{we}$ | 250 m | Hengl, 2018c |
| Clay fraction | % | $BC_{we}$ | 250 m | Hengl, 2018a |
| Bulk density | g/cm$^3$ | $BC_{we}$ | 250 m | Hengl, 2018d |
| Coarse fragment volume | % | $BC_{we}$ | 250 m | Hengl, 2018b |
| Parent material acid class | class | $BC_{we}$ | 250 m | CLBBR, 1996; SLCWG, 2010 |
| Drainage class | class | $BC_{we}$ | 250 m | CLBBR, 1996; SLCWG, 2010 |
| Runoff (Q) | mm yr$^{-1}$ | $ANC_{le,crit}$ | 0.05° x 0.1° | Reinds et al., 2015 |
| Vegetation | | | | |
| Tree species composition | % | $Bc_{up}$, $N_{up}$ | 250 m | Beaudoin et al., 2014 |
| Biomass | Mg ha$^{-1}$ | | | |
| Harvestable boundaries | km$^2$ | $Bc_{up}$, $N_{up}$ | 250 m | Dymond et al., 2010 |
| Tree chemistry database (U.S.) | % Ca, Mg, K, N | $Bc_{up}$, $N_{up}$ | - | Pardo et al., 2005 |
| Tree chemistry database (Can.) | % Ca, Mg, K, N | $Bc_{up}$, $N_{up}$ | - | Paré et al., 2013 |
| Land cover (2010) | class | Limiting extent | 250 m | (CEC, 2018) |
| Agricultural ecumene (2016) | class | Limiting extent | 5 km | Statistics Canada, 2017 |
| Ecozones | class | Limiting extent, summary statistics | 1:7.5 million | Agriculture and Agri-Food Canada, 2013 |
| Deposition | | | | |
| Base cation deposition (2010, 2016) | eq ha$^{-1}$ yr$^{-1}$ | $ANC_{le,crit}$ | 12 km | (Galmarini et al., 2021) |
| Total mean S and N deposition | eq ha$^{-1}$ yr$^{-1}$ | Exceedance | 10 km | (Moran et al., 2024b, a) |



| (2014–2016) | | | | |
| --- | --- | --- | --- | --- |
| Canadian Protected and Conserved Areas Database | class | Identifying areas of special interest | Various | (ECCC, 2023b) |

## 2.4 Base cation weathering

Generalised base cation weathering $BC_{we}$ (i.e., calcium, magnesium, potassium and sodium) was mapped using the soil type–texture approximation method, which assigns a base cation weathering class ($BC_{w0}$) based on soil characteristics (organic matter, sand, and clay percentage) and parent material acid class (see Eq. 6). Weathering is modified by ambient temperature T, where A is the Arrhenius pre-exponential factor (3600 K), a temperature coefficient for soil weathering (de Vries et al., 1992; CLRTAP, 2015). To address issues with resolution and continuity across provinces, high-resolution (global 250 m) predicted soil maps from the OpenLandMap.org project were used for the following input variables: bulk density ($\rho$), organic carbon, coarse fragment volume (CF), and sand and clay composition (see Table 1). One of the assumptions of the SMB model is that the soil compartment is homogeneous; therefore, a weighted average for soil texture was developed based on layer depth, total depth (D), and corrections based on coarse fragment volume, percent organic matter, and bulk density. Percent organic matter (OM) was obtained by dividing organic carbon (in $\times 5$ g kg$^{-1}$) by 2 (as recommended by Hengl & Wheeler, 2018; Pribyl, 2010).

$$BC_{we} = \left(\frac{\rho_{soil}}{\rho_{H_2O}}\right) D \left(1 - \frac{CF}{100}\right)\left(1 - \frac{OM}{100}\right)\left(BC_{w0} - 0.5\right) * 10^{\left(\frac{A}{281} - \frac{A}{273+T}\right)} \qquad (6)$$

A second assumption is that the profile depth (D) is limited to the root zone, which was set to a maximum of 50 cm for forest soils and 30 cm for other land cover types such as shrubland, grassland and tundra. Soil depth was further limited by an absolute-depth-to-bedrock global modelled map (Hengl, 2017; Shangguan et al., 2017) in case bedrock was < 50 cm. Base cation weathering omitting sodium ($Bc_{we}$) required for the calculation of $ANC_{le,crit}$ (Eq. 2) was scaled by 0.8 after CLRTAP (2015).

## 2.5 Base cation deposition

In the absence of modelled $Bc_{dep}$ data, previous Canadian mapping studies have employed a single value, or coarsely interpolated from limited Canadian Air and Precipitation Monitoring Network (CAPMoN) stations from 1994–1998 (Aklilu et al., 2022; Carou et al., 2008; Ouimet et al., 2006). Critical loads estimates for Canada by Reinds et al. (2015) used coarse modelled global Ca deposition (Tegen and Fung, 1995) based on soil Ca content (Bouwman et al., 2002) and estimated the other ions by regression. To address the gaps in data availability and spatial distribution, $Bc_{dep}$ in this study was sourced from modelled estimates produced with the Global Environmental Multiscale–Modelling Air-quality and CHemistry (GEM-MACH) model at 12-km horizontal grid spacing for the air quality multi-model comparison project AQMEII4 (Galmarini et al., 2021). Two different GEM-MACH



configurations, a version with detailed parameterizations and a second version with some simplified
parameterizations used for operational air-quality forecast simulations, estimated wet and dry non-sea-salt Bc$_{dep}$ for
North America. Gridded annual deposition fields for two periods, 2010 and 2016, were obtained. Ideally, emissions
data sources used for S and N deposition and Bc$_{dep}$ would be the same; however, Bc$_{dep}$ is often not evaluated, and the
version of the emissions inventories used for S and N deposition did not include Bc$_{dep}$. Comparison of modelled wet
Bc$_{dep}$ to measured wet Bc$_{dep}$ data from 33 Canadian Air and Precipitation Monitoring Network (CAPMoN)
precipitation-chemistry stations (Feng et al., 2021) and 87 U.S. National Atmospheric Deposition Monitoring
(NADP) precipitation-chemistry stations (NADP, 2023) within 300 km of the Canada-U.S. border showed that
modelled Bc$_{dep}$ data were underestimated in each model configuration and year by an average factor of 15, though
the correlation was relatively high (Figure 3). A Bc$_{dep}$ input map was prepared by averaging (wet plus dry) Bc$_{dep}$
across the two model runs and two years, scaling up by 15 (after Figure 3), and resampling to the 250 m soil grid
using bilinear interpolation.

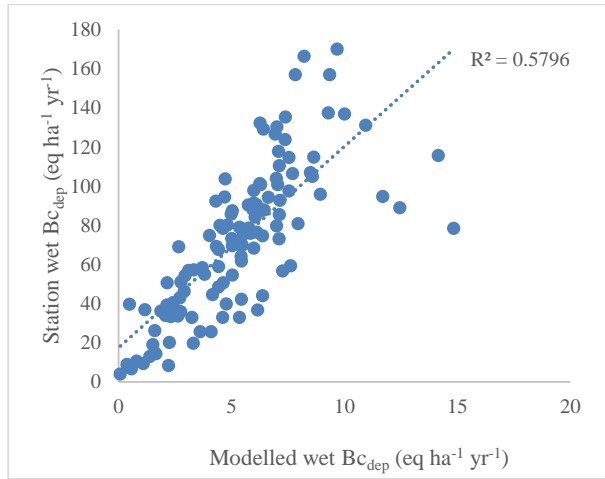


**Figure 3: Modelled annual wet non-sea salt Bc$_{dep}$ (Ca + Mg + K) versus measured annual Bc$_{dep}$ at CAPMoN and NADP
stations (NADP stations limited to those within 300 m of the Canada-U.S. border). Values are averaged across two years
(2010 and 2016) and two model configurations. Marine station sites were corrected for sea salt contributions.**

The modelled Bc$_{dep}$ and station observations include anthropogenic input, but the Bc$_{dep}$ input to the SMB model is
meant to reflect long-term non-anthropogenic sources of base cations. However, large point sources of Bc$_{dep}$ such as
the AOSR are a feature of some Canadian regions, and their impact should not be overlooked in critical load
assessments. To demonstrate the relative impact of anthropogenic sources on Canadian critical loads estimates, two
scenarios were assessed, one including anthropogenic Bc$_{dep}$ and another that attempted to smooth out anthropogenic
"hot spots".

To reduce the influence of anthropogenic point sources, a smoothing filter was applied using the SAGA GIS module
DTM Filter to identify local areas of locally intensified Bc$_{dep}$. Areas of Bc$_{dep}$ above a 30% increase relative to a 20-
grid radius (approximately 50 km) were removed and infilled from their edges using inverse distance weighted



interpolation. Note that forest fire emissions may be substantial and appear as $Bc_{dep}$ hot spots; for this application of
the SMB, we have not added a forest fire term to the base cation budget because of the difficulty of accounting
forest fire loss over the entire country.

**2.6 Soil runoff**


Soil runoff was obtained from the hydrological model MetHyd (Bonten et al., 2016) following Reinds et al. (2015).
The data were resampled from the original resolution of 0.1 x 0.05° to 250 m and gaps were infilled from the edges.
A minimum Q was assigned (10 $m^3$ $ha^{-1}$ $yr^{-1}$) for broad regions where the coarse input soil map (FAO-UNESCO,
2003) used for hydrological modelling did not identify soil (i.e., exposed bedrock), but the high-resolution soil depth
and texture maps used for critical loads did identify soil.

**2.7 Gibbsite equilibrium constant**


The gibbsite equilibrium constant ($K_{gibb}$) describes the relationship between free (or unbound) aluminum
concentration and pH in the soil solution. As free aluminum concentrations are generally lower in the upper organic
horizons, observed ranges based on the organic matter content of the soil may be used to assign a $K_{gibb}$ value. Soils
with organic matter less than 5% were assigned a value of 950 $m^6$ $eq^{-2}$, soils with 5–15% organic matter were
assigned a lower value of 300 $m^6$ $eq^{-2}$ $yr^{-1}$, and soils ranging from 15–30% organic matter were assigned a value of
100 $m^6$ $eq^{-2}$ (after CLRTAP, 2015).

**2.8 Chemical criterion for damage**


The critical base-cation-to-aluminum ratio ($Bc/Al_{crit}$) is the most widely used threshold, indicating damage to root
biomass. It is a simple approach that has been used in past Canadian estimates (e.g., Carou et al., 2008). In general,
it is applied as blanket or default value (e.g., $Bc/Al_{crit}$ = 1) to a range of land cover types (e.g., forest or grassland).
In the current study, a species- and site-specific approach was used to assign damage thresholds for forest
ecosystems based on detailed tree species maps from the 2001 Canadian National Forest Inventory (NFI) (Beaudoin
et al., 2014). Two levels of protection were chosen to illustrate the difference between 20% acceptable growth
reduction (generally analogous to the default $Bc/Al_{crit}$ = 1) versus a 5% growth reduction (generally analogous to
$Bc/Al_{crit}$ = 10). Dose-response curves for $Bc/Al_{crit}$ and root growth from Sverdrup and Warfvinge (1993) were
matched to species present in the NFI database (Table 2). Values were sorted by the most sensitive species (those
with the lowest $Bc/Al_{crit}$) and given priority for the 250 m grid-cell value. If species-specific composition data for
forests (from Beaudoin et al., 2014) were not available, the $Bc/Al_{crit}$ value was averaged to the genus; if no genus-
level data were available, an average coniferous, deciduous, or mixed forest value was applied. For non-forested
soils, a default value based on a representative species for the land cover type was used (e.g., 4.5 and 0.8 for 5% and
20% protection levels, respectively, for grassland based on the response of *Deschampsia*).




**Table 2: Species-specific Bc/Al$_{crit}$ values for 5% and 20% growth reduction scenarios following Sverdrup & Warfvinge (1993). Genus-level or generalised land cover values were derived from representative species.**

| | Bc/Al$_{crit}$ | |
|---|---|---|
| **Category** | **5%** | **20%** |
| **Species (forest)** | | |
| *Abies balsamea* | 6.0 | 1.1 |
| *Fagus grandifolia* | 1.3 | 0.6 |
| *Picea mariana* | 2.5 | 0.8 |
| *Pseudotsuga menzerii* | 4.0 | 2.0 |
| *Pinus strobus* | 1.5 | 0.5 |
| *Picea engelmannii* | 2.5 | 0.5 |
| *Pinus banksiana* | 3.0 | 1.5 |
| *Acer saccharum* | 1.3 | 0.6 |
| *Alnus glutinosa* | 4.0 | 2.0 |
| *Quercus rubra* | 1.3 | 0.6 |
| *Pinus ponderosa* | 4.5 | 2.0 |
| *Pinus resinosa* | 4.5 | 2.0 |
| *Picea rubens* | 6.0 | 1.2 |
| *Picea sitchensis* | 2.5 | 0.4 |
| *Larix laricina* | 4.0 | 2.0 |
| *Populus tremuloides* | 8.0 | 4.0 |
| *Tsuga heterophylla* | 1.0 | 0.2 |
| *Thuja plicata* | 1.0 | 0.1 |
| *Betula papyrifera* | 4.0 | 2.0 |
| *Picea glauca* | 2.5 | 0.5 |
| *Betula alleghaniensis* | 4.0 | 2.0 |
| *Betula populifolia* | 4.0 | 2.0 |
| *Picea abies* | 6.0 | 1.2 |
| *Pinus sylvestris* | 3.0 | 1.2 |
| | | |
| **Genus (forests)** | | |
| Abies | 6.0 | 1.1 |
| Acer | 1.3 | 0.6 |
| Alnus | 4.0 | 2.0 |
| Betula | 4.0 | 2.0 |
| Fagus | 1.3 | 0.6 |
| Larix | 4.0 | 2.0 |
| Picea | 2.5 | 0.8 |
| Pinus | 3.0 | 1.5 |
| Populus | 8.0 | 4.0 |
| Pseudotsuga | 4.0 | 2.0 |
| Quercus | 1.3 | 0.6 |
| Thuja | 1.0 | 0.1 |
| Tsuga | 1.0 | 0.2 |
| | | |
| **Generalised forest** | | |
| Deciduous | 4.0 | 2.0 |
| Coniferous | 3.0 | 1.2 |
| Mixed | 3.0 | 1.2 |
| | | |
| **Generalised land covers** | | |
| Grassland | 4.5 | 0.8 |
| Scrubland | 2.8 | 0.6 |
| Tundra | 2.9 | 0.7 |




**2.9 Base cation and nitrogen uptake**

A species- and site-specific approach was also implemented to determine the net removal of nutrients (Ca, Mg, K, N) through tree harvesting from forest ecosystems. Base cation uptake ($Bc_{up}$) and N uptake ($N_{up}$) were estimated for forest soils by assuming stem-only removal; site-specific stand bark and trunk biomass estimates (Beaudoin et al., 2014) were multiplied by average trunk- and bark-specific nutrient and base cation concentration data from the tree chemistry databases for each species present. Two 'tree chemistry' databases were merged to include as many tree species as possible (U.S. data: Pardo et al., 2005; Canadian data: Paré et al., 2013); duplicate studies were removed from the merged database and species data were averaged across studies. A simplifying assumption was made that stand biomass was related to the species composition (i.e., the dominant tree species in a stand is also the dominant contributor to biomass). The nutrient uptake maps were restricted to harvestable forest areas as delineated by Dymond et al. (2010) and all other regions were set to 0. Nutrient uptake of other land types (e.g., grasslands) was considered negligible since grazing takes place primarily in agricultural regions, which have been broadly masked out. Since $Bc_{up}$ cannot exceed inputs from deposition, weathering, and losses from leaching, a scaling factor was used to constrain base cation uptake between its maximum (that is, deposition + weathering – leaching) and a minimum calcium leaching value. The same scaling factor was applied to $N_{up}$.

**2.10 Denitrification fraction**

The soil denitrification fraction ($f_{de}$) is generally related to soil drainage (CLRTAP, 2015); classes ranging from excessive to very poor drainage were assigned using the Canada-wide Canadian Soil Information Service (CanSIS) databases v2.2 (CLBBR, 1996) and v.3.2 (SLCWG, 2010) (Table 3). In cases of overlapping polygons from the two databases, boundary and classification priority was given to the most recent database version before rasterization.

**Table 3: Denitrification fraction ($f_{de}$) values (adapted from CLRTAP, 2015) and their corresponding drainage classifications in versions 2.2 and 3.2 of the Canadian Soil Information Service database.**

| Drainage | $f_{de}$ | V2.2 | V3.2 |
|---|---|---|---|
| Excessive | 0 | E/R | VR/R |
| Good | 0.1 | W | W |
| Moderate | 0.2 | M | MW |
| Imperfect | 0.4 | I | I |
| Poor | 0.7 | P | P |
| Very poor | 0.8 | V | VP |

**2.11 Deposition and exceedance**

Exceedances for both acidity and nutrient nitrogen were calculated against total deposition maps of annual total S and N, which were sourced from GEM-MACH model output at 10 km horizontal grid spacing (GEM-MACH v3.1.1.0, RAQDPS version 023) (Moran et al., 2024a, b). A three-year (2014–2016) annual average was taken to reduce inter-annual variability in deposition, where input emissions based on annual emissions inventories specific





to each of these three years were used for the three annual runs. Note that Moran et al. (2024b) have presented
detailed evaluations of some components of these deposition estimates, specifically ambient concentration (as a
proxy for dry deposition) and wet deposition of $SO_2$ and particle sulphate (p-$SO_4$), $HNO_3$ and p-$NO_3$, and $NH_3$ and
p-$NH_4$, that suggest that they are robust.

Exceedances of critical load for both acidity and nutrient nitrogen (on a 250 m grid) were summarized to the 10 km
deposition grid using Average Accumulated Exceedance (AAE), which is an area-weighted average that considers
ecosystem coverage within each grid cell to derive the average of the summed exceedance; this addresses issues
with sparse coverage and considers all ecosystems within the grid (Posch et al., 1999). The CPCAD was used to
identify areas in exceedance that may be of particular concern to policymakers (ECCC, 2023b). The database,
assembled in support of Canada's reporting on Canadian Environmental Sustainability Indicators and the UN
Convention on Biological Diversity (among other initiatives), identifies Protected Areas (PA) such as national and
provincial parks as well as Other Effective area-based Conservation Measures (OECM). Interim areas were
included in expectation of their formal establishment. Areas that fell entirely within the agricultural ecumene were
removed, but areas that straddled the ecumene were retained. Areas were counted as in exceedance if any part of the
area experienced exceedance at the 250 m resolution.

The exceedance calculations used for acidity employed the methodology described by Posch et al. (2015), where the
critical load function (Figure 2) was divided into five regions, and a different formula for exceedance was used for
each region. Five inputs for each 250 m grid cell were required for these calculations: the S and N total deposition
pair plus $CL_{max}S$, $CL_{min}N$, and $CL_{max}N$ values. For S and N total deposition pairs falling into four of the regions, the
exceedance value will be positive (i.e., in exceedance) and its magnitude indicates how great the S and N acidic
deposition at the location is above the critical load for acidity. For the Region 0, the exceedance value will be
negative (i.e., not in exceedance) and its magnitude will give how far the S and N acidic deposition is below the
critical load for acidity. Calculation of nutrient N exceedance was simply the difference between $N_{dep}$ and $CL_{nut}N$.
**3 Results**
**3.1 Base cation weathering**
The estimate $BC_{we}$ was very low (below 100 eq ha$^{-1}$ yr$^{-1}$) for nearly all regions north of 60°N latitude, and low
(below 200 eq ha$^{-1}$ yr$^{-1}$) for many northern regions south of 60°N latitude (Figure 4A). Higher $BC_{we}$ (above 500 eq
ha$^{-1}$ yr$^{-1}$) was predicted for the calcareous and deep soils of the Prairies and southern Ontario adjacent to agricultural
regions (i.e. the mean Prairie average for natural and semi-natural soils was 714 eq ha$^{-1}$ yr$^{-1}$), although most of these
ecozones are excluded as part of the agricultural ecumene (Table 4). Average $BC_{we}$ for the Arctic ecozones was <
50 eq ha$^{-1}$ yr$^{-1}$, in contrast with $BC_{we}$ > 700 for Mixed Wood Plain and Prairie ecozones. Similarly, provincial
averages were lowest for Nunavut and highest for Saskatchewan (Table 4). Base cation weathering without



temperature correction (Figure 4B, mean value of 570 eq ha$^{-1}$ yr$^{-1}$) illustrates the strong effect temperature has on
limiting BC$_{we}$ in most of the country (average 173 eq ha$^{-1}$ yr$^{-1}$), particularly Arctic and mountainous regions.

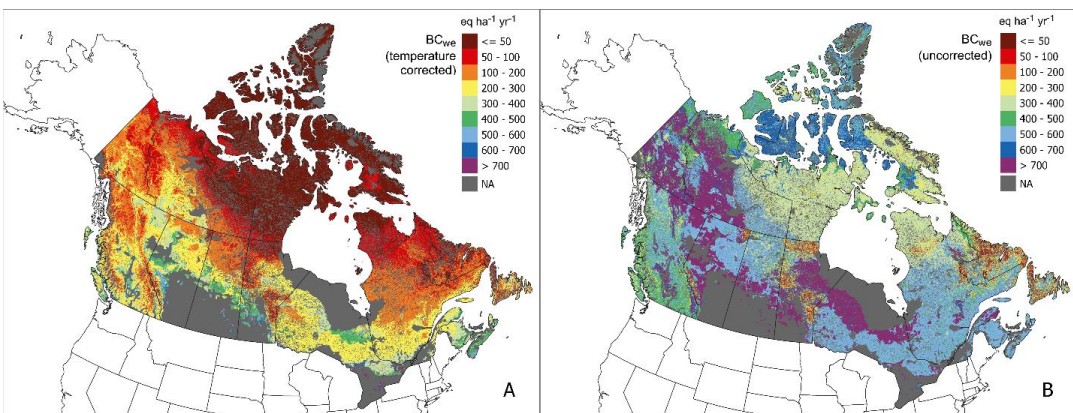


**Figure 4: Base cation weathering rate (Ca+Mg+K+Na) with temperature correction (A) and without (B). The weathering**
**rate was estimated using a soil texture approximation method with sand, clay, and parent material acid class modified by**
**depth (see Section 2.4).**
**Table 4: Ecozone and provincial mean values for inputs and outputs of the Simple Mass Balance model, including base**
**cation weathering (BC$_{we}$), smoothed base cation deposition (Bc$_{dep}$), base cation uptake (Bc$_{up}$), nitrogen uptake (N$_{up}$),**
**critical base-cation-to-aluminum ratio (Bc/Al$_{crit}$) under 5% and 20% growth reduction scenarios, average sulphur**
**deposition (DepS) and nitrogen deposition (DepN) 2014 - 2017), maximum critical load of sulphur (CL$_{max}$S), maximum**
**critical load of nitrogen (CL$_{max}$N), minimum nitrogen critical load (CL$_{min}$N) and critical load of nutrient nitrogen**
**(CL$_{nut}$N). Units are in eq ha$^{-1}$ yr$^{-1}$ except for Bc/Al$_{crit}$ which is a unitless ratio. The critical loads presented in the table**
**were calculated using the 5% Bc/Al$_{crit}$ and the smoothed Bc$_{dep}$. Note that values represent coverage over eligible soils (e.g.**
**excluding agricultural areas and organic soils).**

| Ecozone | BC$_{we}$ | Bc$_{dep}$ | Bc$_{up}$ | N$_{up}$ | Bc/Al$_{crit}$ 5% | Bc/Al$_{crit}$ 20% | DepS | DepN | CL$_{max}$S | CL$_{max}$N | CL$_{min}$N | CL$_{nut}$N |
|---|---|---|---|---|---|---|---|---|---|---|---|---|
| Arctic Cordillera | 40 | 5 | < 1 | < 1 | 5.2 | 1.2 | 8 | 21 | 82 | 88 | 36 | 173 |
| Atlantic Maritime | 353 | 89 | 32 | 37 | 5.7 | 1.1 | 57 | 240 | 615 | 551 | 36 | 234 |
| Boreal Cordillera | 174 | 19 | 5 | 5 | 3.9 | 1.2 | 10 | 29 | 290 | 274 | 36 | 77 |
| Boreal Plain | 331 | 139 | 27 | 23 | 3.7 | 1.4 | 38 | 172 | 802 | 549 | 36 | 71 |
| Boreal Shield | 229 | 84 | 18 | 23 | 4.0 | 0.9 | 53 | 206 | 512 | 422 | 36 | 147 |
| Hudson Plain | 212 | 56 | < 1 | < 1 | 2.8 | 0.8 | 30 | 111 | 499 | 221 | 36 | 104 |
| Mixedwood Plain | 712 | 180 | < 1 | < 1 | 3.6 | 0.9 | 137 | 712 | 1586 | 1171 | 36 | 145 |
| Montane Cordillera | 240 | 52 | 39 | 42 | 3.6 | 1.2 | 25 | 98 | 447 | 473 | 40 | 164 |
| Northern Arctic | 32 | 7 | < 1 | < 1 | 5.6 | 1.3 | 9 | 20 | 63 | 75 | 41 | 75 |
| Pacific Maritime | 274 | 25 | 78 | 135 | 2.9 | 0.9 | 53 | 172 | 608 | 1281 | 48 | 513 |
| Prairie | 559 | 191 | 13 | 2 | 5.0 | 1.9 | 54 | 423 | 1078 | 893 | 59 | 63 |
| Southern Arctic | 45 | 21 | < 1 | < 1 | 5.6 | 1.3 | 10 | 26 | 112 | 118 | 60 | 65 |



| | | | | | | | | | | | |
|---|---|---|---|---|---|---|---|---|---|---|---|
| Taiga Cordillera | 106 | 31 | < 1 | < 1 | 4.3 | 1.0 | 10 | 26 | 218 | 194 | 76 | 66 |
| Taiga Plain | 195 | 51 | 4 | 4 | 3.2 | 1.0 | 13 | 37 | 390 | 246 | 79 | 51 |
| Taiga Shield | 88 | 40 | < 1 | < 1 | 3.5 | 0.8 | 18 | 54 | 227 | 200 | 192 | 110 |
| **Province** | | | | | | | | | | | |
| Alberta | 285 | 133 | 24 | 17 | 3.7 | 1.4 | 35 | 142 | 730 | 512 | 58 | 78 |
| British Columbia | 235 | 37 | 40 | 53 | 3.6 | 1.2 | 26 | 92 | 439 | 551 | 91 | 206 |
| Manitoba | 217 | 86 | 7 | 7 | 2.9 | 0.9 | 41 | 146 | 512 | 338 | 44 | 66 |
| New Brunswick | 344 | 91 | 34 | 41 | 5.8 | 1.1 | 49 | 227 | 595 | 502 | 79 | 243 |
| Newfoundland & Labrador | 110 | 24 | 6 | 7 | 4.6 | 0.9 | 24 | 71 | 217 | 190 | 43 | 223 |
| Nova Scotia | 422 | 92 | 21 | 28 | 5.6 | 1.1 | 68 | 249 | 733 | 652 | 65 | 261 |
| Northwest Territories | 114 | 41 | < 1 | < 1 | 4.0 | 1.0 | 11 | 28 | 254 | 191 | 36 | 49 |
| Nunavut | 34 | 11 | < 1 | < 1 | 5.5 | 1.2 | 9 | 22 | 75 | 87 | 36 | 75 |
| Ontario | 306 | 103 | 23 | 19 | 3.8 | 0.9 | 61 | 289 | 666 | 509 | 66 | 141 |
| Prince Edward Island | 422 | 69 | 19 | 18 | 5.3 | 1.0 | 57 | 209 | 672 | 558 | 66 | 226 |
| Québec | 148 | 46 | 11 | 14 | 4.5 | 1.0 | 38 | 132 | 314 | 299 | 50 | 153 |
| Saskatchewan | 230 | 124 | 12 | 8 | 3.1 | 1.0 | 29 | 128 | 607 | 492 | 49 | 62 |
| Yukon | 148 | 25 | < 1 | < 1 | 3.8 | 1.0 | 10 | 26 | 266 | 233 | 36 | 54 |
| **Canada** | 132 | 52 | 8.2 | 10 | 4.5 | 1.1 | 76 | 22 | 291 | 258 | 48 | 99 |


## 3.2 Base cation deposition

Modelled $Bc_{dep}$ ranged from low (< 25 eq ha$^{-1}$ yr$^{-1}$) in the north to higher values (> 200 eq ha$^{-1}$ yr$^{-1}$) around the
Prairies and the southern regions of the eastern provinces (Figure 5) as well as in Alberta and Saskatchewan (Table
4). Average (smoothed) $Bc_{dep}$ was roughly one-third of $BC_{we}$. Hot spots of $BC_{dep}$ associated with at anthropogenic
point sources (e.g., from mining operations as well as the contribution from the AOSR) were clearly visible in the
unsmoothed map (Figure 5A). The smoothing algorithm (Figure 5B) eliminated most of the effects of point sources,
at the cost of some loss of definition (Canada-wide average of 52 eq ha$^{-1}$ yr$^{-1}$ pre-smoothing and 68 eq ha$^{-1}$ yr$^{-1}$ post-
smoothing). However, it did not completely erase elevated $Bc_{dep}$ in the AOSR; the difference in size between other
point source footprints and the AOSR neccessitated a compromise in filter radius and slope selection.



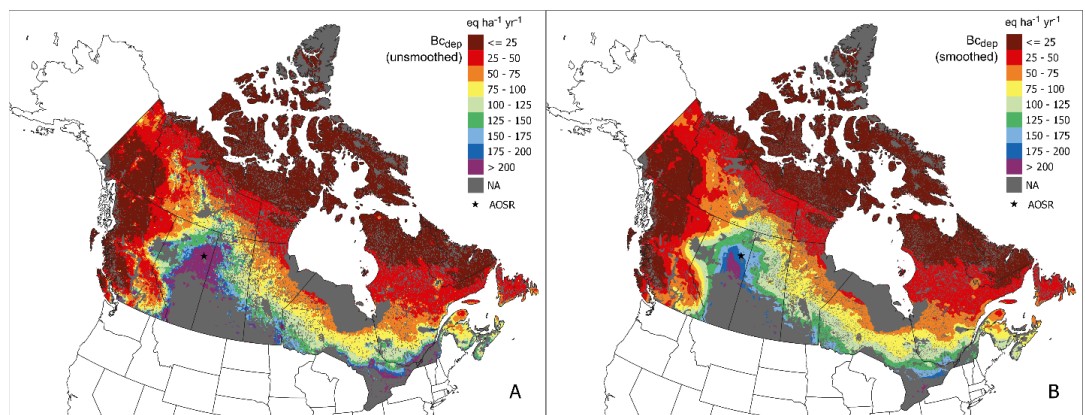


**Figure 5: Non-sea-salt base cation deposition (Ca + Mg + K) with anthropogenic contributions (A) and after a smoothing filter was applied to reduce the effect of anthropogenic point sources (B). The Athabasca Oil Sands Region (AOSR) is identified by a star.**

### 3.3 Base cation and nitrogen uptake

Base cation uptake ranged from < 1 to 545 eq ha$^{-1}$ yr$^{-1}$ and was highest in coastal British Columbia; the Pacific Maritime ecozone had the highest mean $Bc_{up}$ at 79 eq ha$^{-1}$ yr$^{-1}$ (Table 4). Nitrogen uptake was also high in British Columbia and the Pacific Maritime zone (mean $N_{up}$ of 135 eq ha$^{-1}$ yr$^{-1}$) as well as the Montane Cordillera (mean $N_{up}$ of 42 eq ha$^{-1}$ yr$^{-1}$). Regions of elevated $N_{up}$ were seen in eastern Ontario and southern Quebec (Figure 6); these occur on the Boreal Shield ecozone, which is a large ecozone that extends across multiple provinces over which $N_{up}$ varies (but with a mean value of 23 eq ha$^{-1}$ yr$^{-1}$).

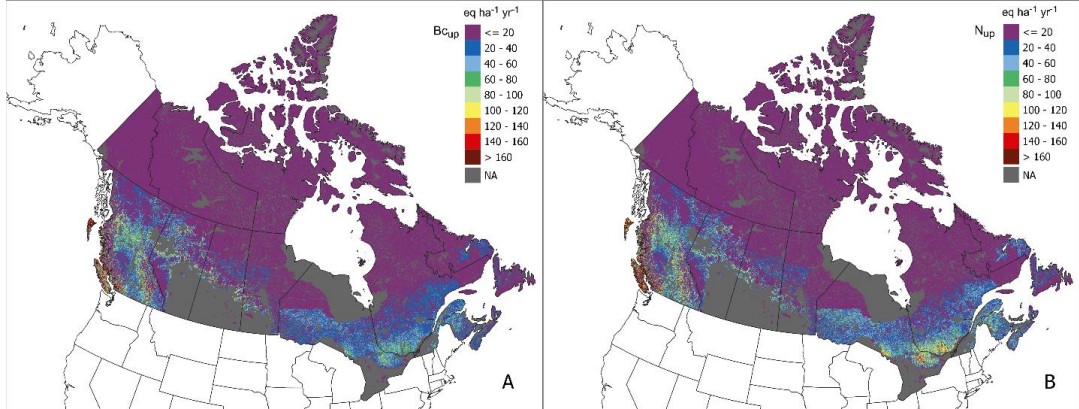

352

**Figure 6: Base cation (Ca+Mg+K) uptake (A) and nitrogen uptake (B;) forested regions limited to harvestable regions (identified by Dymond et al. (2010)). Uptake for non-forested ecosystems was set to 0.**

### 3.4 Critical base-cation-to-aluminum ratio

Almost the entire country fell below a Bc/Al$_{crit}$ ratio of 2 under 20% root biomass growth reduction (Figure 7A). In contrast, a Bc/Al$_{crit}$ ranged from 1–8 (average = 4.4) under the 5% root biomass growth reduction (Figure 7B). The ratio ranged from 3–6 for forests in eastern Canada (A and B ecozones), while ranges for the Boreal Shield ecozone





were 2–4 and coastal forest in British Columbia were slightly higher at 3–4. Semi-natural grassland in the Prairies
were given a ratio of 4.5 based on *Deschampsia*, but many fringe regions of the Prairies are treed and dominated by
*Populus tremuloides*, which had a Bc/Al$_{crit}$ of 8.

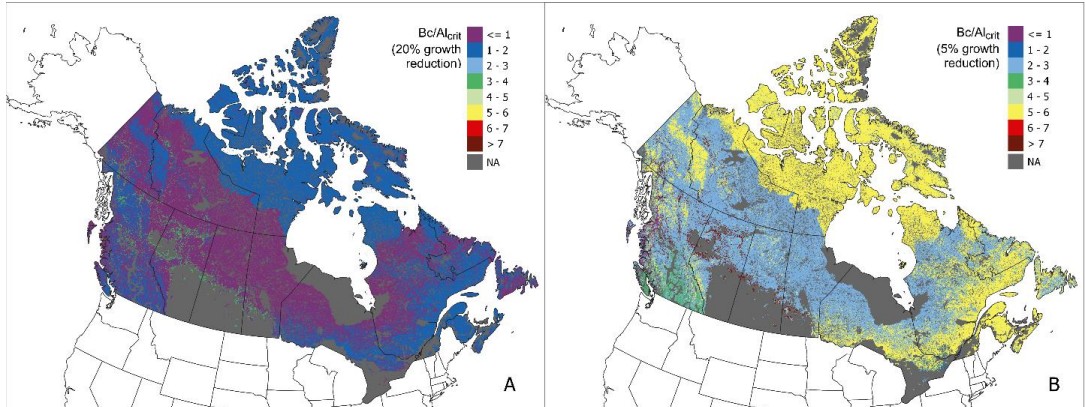


**Figure 7: Critical base-cation-to-aluminum ratio (Bc/Al$_{crit}$) under a 20% growth reduction (A) and a 5% growth**
**reduction (B). Site-specific ratios were selected for each 250 m grid cell for the most sensitive species (or genus or land-**
**cover type if no species data available). Note that while the legends have been matched for comparison, the maximum**
**ratio in the 20% growth reduction map is 4.**
**3.5 Critical loads**
The CL$_{max}$S under the 20% protection level (i.e., allowing more damage) showed low sensitivity (> 1000 eq ha$^{-1}$ yr$^{-1}$
$^1$) to acidic deposition for most regions below 55°N latitude (Figure 8A). In contrast, under the 5% protection level
(Figure 8B), low sensitivity was limited to southern agricultural regions in the Prairies. Lowest CL$_{max}$S and CL$_{max}$N
were found in the Arctic territories (Nunavut, the Northwest Territories, the Yukon; Table 4) and also
Newfoundland and Labrador (Figure 12B). Of the provinces, Quebec had the lowest CL$_{max}$S (314 eq ha$^{-1}$ yr$^{-1}$) and
CL$_{max}$N (299 eq ha$^{-1}$ yr$^{-1}$) (Table 4).

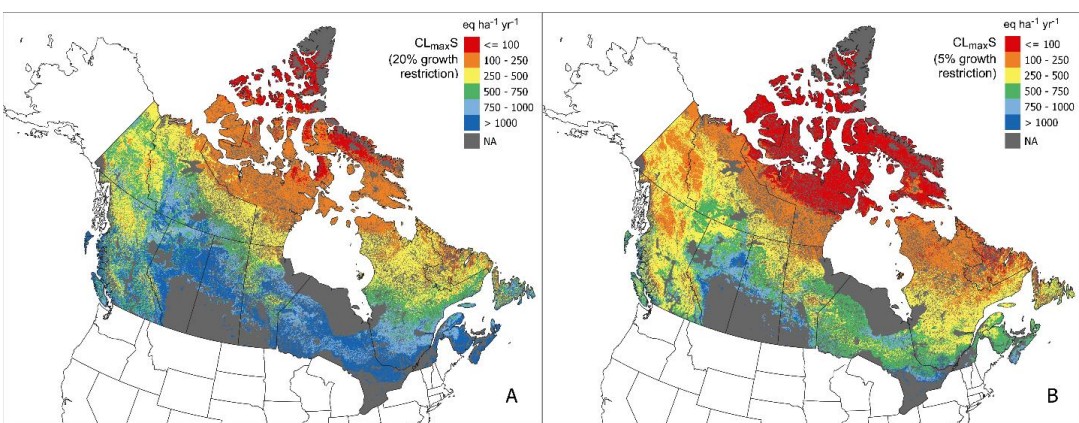


**Figure 8: Maximum sulphur critical load (CL$_{max}$S) at a 20% growth restriction scenario (A) versus a 5% growth**
**restriction scenario (B), using reduced-anthropogenic (i.e., smoothed) Bc$_{dep}$.**



**3.6 Deposition**


Modelled average annual $S_{dep}$ was below 25 eq ha$^{-1}$ yr$^{-1}$ for most of the country above 59°N, as well as the Montaine
Cordillera ecozone that covers much of British Columbia (Figure 9A). Southern Quebec and central Ontario
showed higher annual average values between 50–200 eq ha$^{-1}$ yr$^{-1}$, with some isolated point sources showing $S_{dep}$ in
excess of 500 eq ha$^{-1}$ yr$^{-1}$. Modelled average annual $N_{dep}$ (Figure 9B) exceeded $S_{dep}$ in most parts of the country.
Nitrogen deposition exceeding 500 eq ha$^{-1}$ yr$^{-1}$ was present in northern Ontario and southern Quebec as well as
southern Manitoba and southwestern British Columbia.

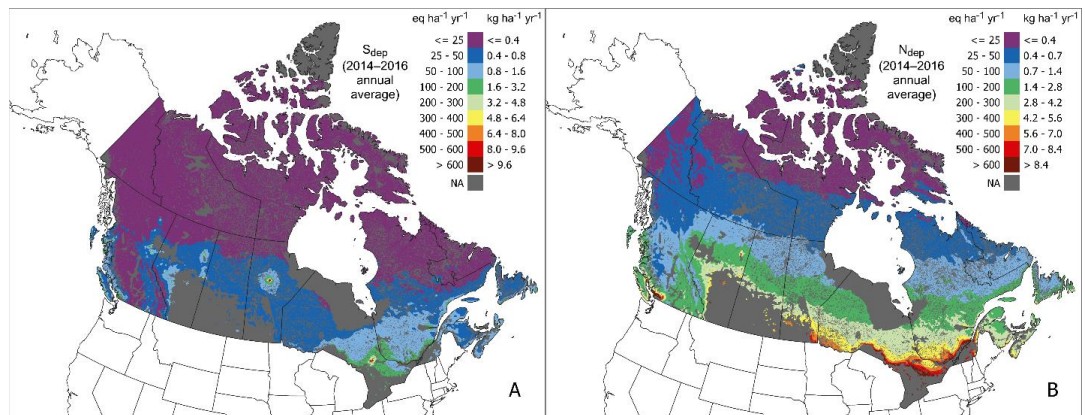


**Figure 9: Modelled total deposition of sulphur ($S_{dep}$, panel A) and nitrogen ($N_{dep}$, panel B) under average annual**
**deposition from 2014–2016. Maps were sourced from GEM-MACH (Moran et al., 2024a, b).**

**3.7 Exceedances**


Widespread but low exceedances of acidity (< 50 eq ha$^{-1}$ yr$^{-1}$) under 2014–2016 deposition were found in regions in
central and southern Quebec, Ontario, Manitoba, Alberta, British Columbia as well as in some regions in Nova
Scotia and Newfoundland, under both protection levels (Figure 10). Further, exceedances above 200 eq ha$^{-1}$ yr$^{-1}$
were predicted in southern Quebec and Ontario, as well as near Winnipeg and Vancouver, under both protection
levels. Exceedances of acidity under 2014–2016 S and N deposition were not generally predicted in the north. The
spatial extent of exceedance was slightly greater under the 5% protection limit as a result of higher $CL_{max}S$ and
$CL_{max}N$, particularly around point sources of S and N, such as the AOSR.

If the $Bc_{dep}$ without smoothing is employed (i.e., the base cation deposition associated with high magnitude
anthropogenic sources is included), exceedances are reduced (see Figure 11(B) and compare to Figure 10(B)). The
$CL_{max}S$ based on anthropogenic-inclusive $Bc_{dep}$ (at 5% protection level, Figure 11A) indicated that $CL_{max}S$ is
elevated in the AOSR in comparison with the smoothed $CL_{max}S$ in Figure 8B.





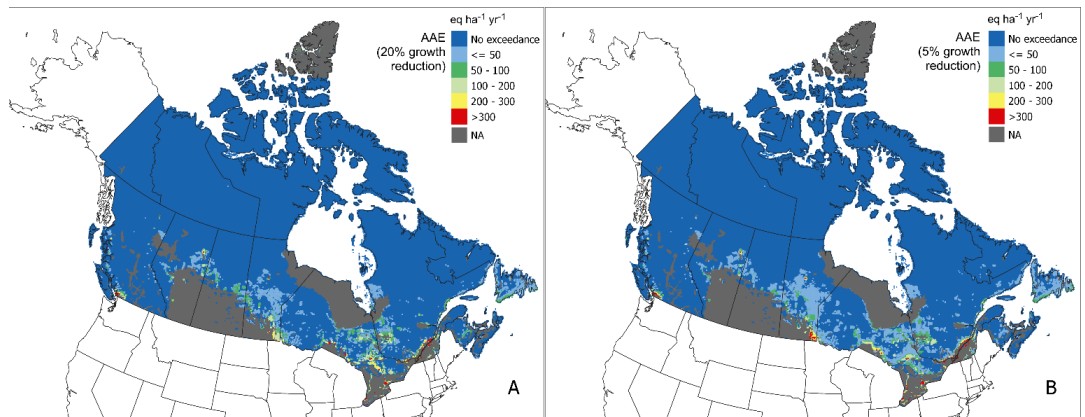

**Figure 10: Average Accumulated Exceedance (AAE) of critical loads of acidity under 2014–2016 sulphur plus nitrogen GEM-MACH modelled deposition. Two growth reduction scenarios are presented: using a chemical criterion representing 20% growth reduction (A) and 5% growth reduction (B).**

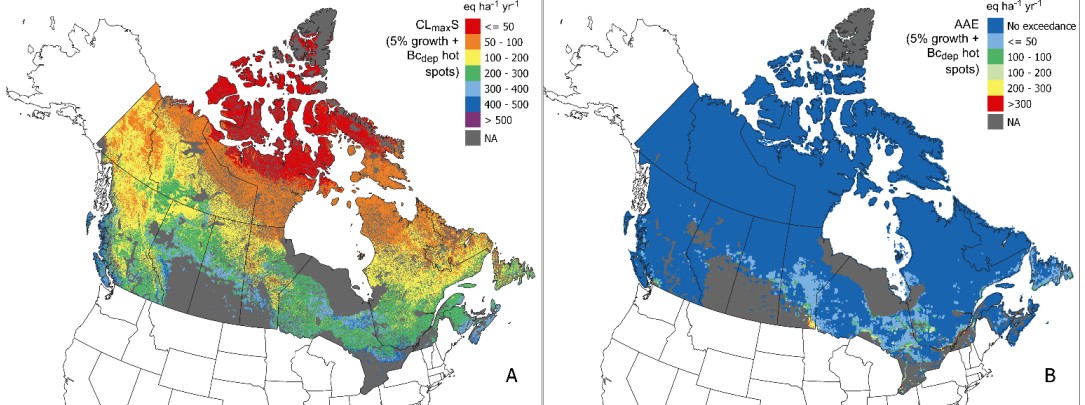

**Figure 11: A scenario including base cation deposition without smoothing, illustrating the impact of hot-spot Bc$_{dep}$ on the maximum critical load of sulphur (CL$_{max}$S) (A) and the Average Accumulated Exceedance (AAE) under 2014–2016 sulphur plus nitrogen GEM-MACH modelled deposition (B).**

For CL$_{nut}$N, central and northern regions of the country were sensitive to nutrient N deposition, particularly pastures, grasslands, scrublands, and sparse forest in and surrounding the Prairies (Figure 12A). Further, very low CL$_{nut}$N (<= 75 eq ha$^{-1}$ yr$^{-1}$ were estimated over the Arctic territories (Table 4) as well as in northern Alberta and the Athabasca Basin in northern Saskatchewan (Figure 12A). Widespread exceedances of CL$_{nut}$N were predicted across most provinces, with generally low AAE (< 50 eq ha$^{-1}$ yr$^{-1}$) extending to just north of 60° latitude, and higher values of 100–200 eq ha$^{-1}$ yr$^{-1}$ were predicted from Alberta east to Quebec (Figure 12B). Some regions adjacent to the agricultural ecumene in the Prairies, southern Ontario, Quebec and the AOSR experienced values above 300 eq ha$^{-1}$ yr$^{-1}$ (Figure 12B).

There were 12,341 sites of interest across Canada (i.e., PA and OECM areas); however, only 8,372 fall within areas assessed in this study (e.g. not within the agricultural ecumene or Hudson Bay Plains ecozone). In total, 10% of



these sites exceeded $CL_{max}S$ under the 5% protection limit (Table 5). This was roughly double the number of sites in
exceedance under the 20% protection limit. By comparison the $Bc_{dep}$ layer with unsmoothed hot spots (i.e. retaining
higher $Bc_{dep}$ close to anthropogenic emissions areas) under the 5% protection limit showed a reduction in total areas
that are in exceedance of acidity critical loads; anthropogenic emissions of base cations reduce the exceedances by
reducing $N_{up}$ values. The number of PA and OECM sites in exceedance of $CL_{nut}N$ was much higher, 70% of total
sites assessed (Table 5).

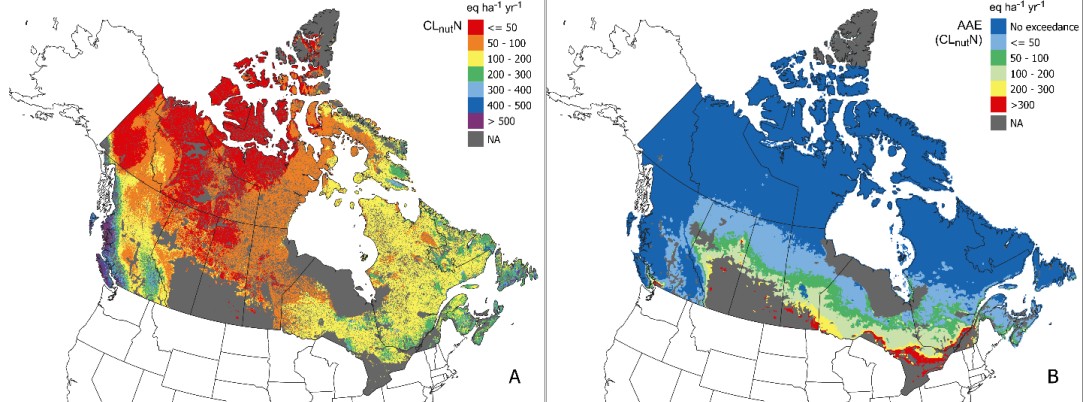


Figure 12: Critical load of nutrient nitrogen using the SMB model (A) and average accumulated exceedance of
nutrient nitrogen (B) estimated under modelled total deposition of nitrogen from 2014–2016.

**Table 5: Exceedance summarized by number of Protected Areas (PA) and Other Effective area-based Conservation**
**Measures (OECM) areas (ECCC, 2023b) experiencing any exceedance. Three exceedance scenarios are presented:**
**Critical load of acidity exceedance at 5% and 20% growth reduction protection levels, unsmoothed base cation deposition**
**under the 5% scenario, and exceedance of nutrient nitrogen ($CL_{nut}N$). Critical loads of acidity and nutrient nitrogen**
**were assessed under a multi-year (2014–2016) average GEM-MACH modelled sulphur and nitrogen total deposition.**

|  | PA | OECM | % Exceeded |
|---|---|---|---|
| Number of sites | 8,205 | 167 | - |
| Exceeded (5% growth reduction) | 793 | 17 | 9 |
| Exceeded (20% growth reduction) | 313 | 10 | 3 |
| Exceeded (5% with hot spots) | 445 | 14 | 5 |
| Exceeded ($CL_{nut}N$) | 5,807 | 85 | 70 |


**4 Discussion**
**4.1 Uncertainties of critical loads of acidity and nutrient nitrogen**
Critical loads of acidity reflect the influence of $BC_{we}$, particularly in the north where cold annual temperatures slow
weathering rates to almost zero. However, areas near the Canada-U.S. border also showed lower $BC_{we}$ rates by 200–
300 eq ha$^{-1}$ yr$^{-1}$ when corrected for temperature (Figure 4). Soil depth remains a poorly mapped parameter that has
significant impact on $BC_{we}$, and it is worth noting that average estimates were based on mapped soil depths (Hengl,



2017), which ranged from 1 cm to a maximum rooting depth of 30 or 50 cm. While comparison between mapped
values and site-level values is difficult (due to methodological differences and spatial representation), there are some
studies which have observational values in representative areas; for example, in northern Saskatchewan, 50% of 107
sites were estimated below 300 eq ha$^{-1}$ yr$^{-1}$, slightly above our mapped estimates of 230 eq ha$^{-1}$ yr$^{-1}$ for (primarily
northern) Saskatchewan (Table 4; Figure 4). Estimates for conifer stands in Québec by Ouimet et al. (2001) were
210 eq ha$^{-1}$ yr$^{-1}$, comparable to the mean 229 eq ha$^{-1}$ yr$^{-1}$ estimated for the Boreal Shield ecozone in our study (Table
4). In British Columbia, Mongeon et al. (2010) found BC$_{we}$ to be 710 eq ha$^{-1}$ yr$^{-1}$, much greater than the 235 eq ha$^{-1}$
yr$^{-1}$ estimated in our study for the Pacific Maritime ecozone. Koseva et al., (2010) estimated BC$_{we}$ at 10 sites in
Ontario primarily in the Mixedwood Plains ecozone at 628 eq ha$^{-1}$ yr$^{-1}$ (compared to 306 eq ha$^{-1}$ yr$^{-1}$ over the
Mixedwood Plains in our study). Moreover, Koseva et al. suggest that the soil-texture approximation method (as
used in our study) under-estimates BC$_{we}$ in comparison to the better-preforming PROFILE model. Assessments of
uncertainty in critical load estimates recognize BC$_{we}$ as the primary driver of uncertainty (Li and Mcnulty, 2007;
Skeffington et al., 2006) and, as such, observational data and PROFILE-modelled site data to constrain weathering
rates would greatly improve critical load estimates.

While the inclusion of a modelled Bc$_{dep}$ map represents an improvement over previous Canadian critical load map
projects, several factors likely contribute to the Bc$_{dep}$ modelled negative bias (which has appeared in other
publications, such as Makar et al., 2018), and may relate to how emissions processing has been carried out for air-
quality models in North America. While anthropogenic emissions inventories include estimates of PM$_{2.5}$, PM$_{10}$ and
PM$_{total}$ mass emissions, usually only PM$_{2.5}$ and PM$_{10}$ emissions are used in determination of model input emissions.
However, substantial emitted base cation mass may reside in the larger size fractions (between the mass included
within PM$_{10}$ and the PM$_{total}$). The model version and emissions inventory data used in the base cation deposition
estimates of AQMEII4 included only emissions up to 10 µm diameter, as did work examining emissions from
multiple sources of primary particulate matter (Boutzis et al., 2020). Subsequent work using observations from the
Canadian Oil Sands and reviewing other sources of data subsequent to Boutzis et al. (2020) and Galmarini et al.
(2021) suggest that many of the same sources of anthropogenic particulate matter emissions include emitted
particles between 10 and 40 µm diameter, the mass of which adds an additional 66% relative to the PM$_{2.5}$ to PM$_{10}$
"coarse mode" emitted mass. For forest fire emissions, this additional mass is much larger. The wildfire particulate
matter size distributions of Radke et al. (1988; 1990) used to estimate mass up to PM$_{10}$ in Boutzis et al. (2020) show
that the emitted particle mass between 10 and 40 µm diameter is 7.26× that emitted between PM$_{2.5}$ and PM$_{10}$.
Approximately 9.7% of this particle mass is composed of base cations (e.g., Table S5 of Chen et al., 2019). A third
factor is another natural emissions source, aeolian or wind-blown dust emissions (e.g., Bullard et al., 2016; Park et
al., 2010), which was not included in the AQMEII4 simulations. These (traditionally missing) sources of base
cation mass in air-quality models likely contribute to the substantial negative bias noted here. Nevertheless,
regression in Figure 3 suggests that the spatial distribution of base cations emissions and deposition from Galmarini
et al. (2021) is reasonable, and we have used the relationship between modelled and observed values to provide
corrected estimates of Bc$_{dep}$.




The conservative 5% protection level set for the $Bc/Al_{crit}$ is favoured by the authors of the current work for critical
loads estimates, which affords greater ecosystem protection consistent with studies using $Bc/Al > 1$ (e.g. McDonnell
et al., 2023; Mongeon et al., 2010; Ouimet et al., 2006). Historically, when acidic deposition was higher than at
present, a 20% growth reduction was a reasonable target. However, under decreasing emissions and deposition, as
well as acceptable impacts to wood production, carbon storage, and ecosystem health there is greater certainty in
ecosystem protection under the 5% protection level. It should be noted that the level of protection is an ethical
choice regarding how much should be protected, rather than a sensitivity, and taking the most sensitive species
through the $Bc/Al_{crit}$ selection process ensures the highest possible protection based on species-specific dose-
response curves. Note, however, that changes to forest health and climate may also induce pressures that are not
captured in the selection of the $Bc/Al_{crit}$ from the studies described in Sverdrup & Warfvinge (1993).
Low $CL_{nut}N$ in the Arctic was driven by very low Q values on thin barren land covers. In contrast, areas with high
Q were found to result in high $CL_{nut}N$; as previously suggested by Reinds et al. (2015), a critical flux rather than
concentration may provide more reliable critical loads in regions with elevated precipitation such as the Pacific
Maritime ecozone in British Columbia.

The omission of wetlands, which cover an estimated 13% of land in Canada, from acidity and nutrient N critical
loads represents a gap in terrestrial (and aquatic) ecosystem protection. Although there are modifications to the
SMB model that address critical loads for wetlands, this study was limited by the availability of a suitable national
wetlands classification map. Future studies may address this data gap as wetland classification products become
available.

**4.2 Exceedances of critical loads**

Historically, forests in eastern Canada were regarded as the region most susceptible to acidification due to their
underlying geology, shallow soil type, vegetation, and elevated acidic deposition from domestic and transboundary
air pollution. This study adds to the body of literature supporting recent studies in both terrestrial and aquatic
critical loads (e.g., Makar et al., 2018; Cathcart et al., 2016; Williston et al., 2016; Mongeon et al., 2010; Whitfield
et al., 2010), showing likely exceedance of critical loads of acidity in central and western Canada (i.e., in regions
such as Alberta, Saskatchewan and British Columbia). The prevalence of our predicted widespread exceedances in
Manitoba (Figure 10) may reflect low mineral soil depth, as organic soil dominates this part of the country. Further,
point sources (generally large mining or smelting operations) remain a concern (e.g., in southern Manitoba, the
AOSR, and southern British Columbia) with regard to sharply elevated local exceedance, which may be temporally
mitigated by elevated $Bc_{dep}$ from co-located dust emissions sources. Additionally, high $Bc_{dep}$ can have an
alkalinizing impact on ecosystems. In China, where elevated $Bc_{dep}$ emissions from industrialization have
historically mitigated the effects of acidic deposition in many regions, successful particle emissions mitigation
strategies have reduced $Bc_{dep}$ in recent years (as S and N deposition have declined), resulting in increased critical
load exceedance (Zhao et al., 2021). However, the steady-state assumptions of the SMB require non-anthropogenic



Bc$_{dep}$, since they must reflect long-term conditions, and base cation emissions cannot be reliably coupled with
changes to those of S and N and should be considered separately.

Widspread CL$_{nut}$N exceedance (found in the majority of the PA and OECM sites assessed) suggests that nutrient N
may present a risk to biodiversity at many sites under protective measures. While some empirical studies of nutrient
N have been done in Canada, a large knowledge gap exists for many Canadian ecosystems regarding the effect of
nutrient nitrogen and their critical loads.  Some work has developed on Jack Pine and northern ecosystems;
Vandinthner suggested that across Jack pine-dominant forests surrounding the AOSR, the biodiversity-based
empirical critical load of nutrient N was 5.6 kg ha$^{-1}$ yr$^{-1}$ (400 eq ha$^{-1}$ yr$^{-1}$; Vandinther and Aherne, 2023a) which
is above the maximum CL$_{nut}$N calculated in this study within 200 km of the AOSR (216 eq ha$^{-1}$ yr$^{-1}$). Further, in low
deposition 'background' regions a biodiversity-based empirical critical load of 1.4–3.15 kg ha$^{-1}$ yr$^{-1}$ (100 – 225 eq
ha$^{-1}$ yr$^{-1}$) was found to protect lichen communities and other N-sensitive species in Jack pine forests across
Northwestern Canada (Vandinther and Aherne, 2023b); these are again higher compared to mean values in this
study (e.g. for the Boreal Plain, 76 eq ha$^{-1}$ yr$^{-1}$).  Empirical critical loads developed for ecoregions in Northern
Saskatchewan (Murray et al., 2017) fall into a range of 88 – 123 eq ha$^{-1}$ yr$^{-1}$, again higher than values suggested by
this study (e.g. 62 eq ha$^{-1}$ yr$^{-1}$ in Saskatchewan).  While the spatial pattern of CL$_{nut}$N exceedances does not generally
follow exceedances of critical loads of acidity, some areas (including PA and OECM sites) in central Canada were
estimated to be in exceedance of both critical loads of acidity and nutrient N, suggesting that this region may be of
particular concern.
**5 Conclusions**
This study mapped critical loads of acidity and nutrient nitrogen for terrestrial ecosystems the using the steady-state
SMB model.  The modelling approach used (a) high-resolution national maps of soils, meteorology, and forest
composition, (b) high-resolution modelled Canada-wide Bc$_{dep}$, and (c) species-specific chemical criteria for damage.
The resulting national critical loads of acidity and nutrient N for Canadian terrestrial ecosystems were mapped at a
250 m resolution.  The influence of different levels of protection and Bc$_{dep}$ models to several parameters was also
explored, including two vegetation protection levels (5% and 20% root biomass growth reduction scenarios) and
anthropogenic base cation deposition "hot spots".

Terrestrial ecosystems in Canada continue to receive acidic deposition in excess of their critical loads for both
acidity and nutrient N under modelled (2014–2016) total S and N deposition in areas of both eastern and western
Canada.  These areas include several major point emissions sources including the Alberta Oil Sands Region.
Further, exceedance was predicted at 10% (acidity) and 70% (nutrient nitrogen) of the assessed sites (PA and
OECM) where preserving biodiversity is a national policy goal, suggesting that current levels of N deposition may
be affecting a large majority of these ecologically important sites.  Soil recovery from acidic deposition is a slow
process that may take decades or even centuries to reach pre-acidification levels, which cannot begin until
deposition falls below critical loads.  Parameterization of the SMB model specifically for Canadian ecosystems is a



step forward in refining Canadian terrestrial critical loads, and the maps produced by this study are a valuable tool in
identifying and assessing regions sensitive to acidic deposition and nutrient N deposition, as well they provide a
foundation for more refined provincial estimates.
**CRediT authorship contribution statement**
**H. Cathcart:** Conceptualization, Data curation, Investigation, Methodology, Formal analysis, Visualization, Writing
– original draft, Writing – review & editing. **J. Aherne:** Formal analysis, Methodology, Writing – review &
editing. **M.D. Moran:** Data curation, Investigation, Methodology, Writing – original draft, Writing – review &
editing. **V. Savic-Jovcic:** Data curation, Investigation. **P.A. Makar:** Investigation, Methodology, Writing – original
draft, Writing – review & editing. **A.D Cole:** Writing – review & editing.
**Competing interests**
The authors declare that they have no conflict of interest.
**Data availability**
Raster files of critical load maps ($Cl_{max}S$, $CL_{max}N$, $CL_{min}N$, $CL_{nut}N$) will be made available on the Government of
Canada's Open Data Portal under Environment and Climate Change Canada's records
(https://open.canada.ca/data/organization/ec).
**Acknowledgements**
This study was funded by Environment and Climate Change Canada. The authors wish to acknowledge Max Posch
for his provision of (and guidance regarding) soil water runoff (Q) estimates, and the AQMEII4 project for
emissions data leading to GEM-MACH maps of base cation deposition.

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
