# Peer review of "Estimates of critical loads and exceedances of acidity and nutrient nitrogen for mineral soils in Canada for 2014–2016 average annual sulphur and nitrogen atmospheric deposition"

_EGUsphere, 2024_

## Referee Comment (RC2)

Review of MS EGUsphere-2024-2371:

**"Estimation of critical loads and exceedances of acidity and nutrient nitrogen for mineral soils in Canada for 2014-2016 average annual sulphur and nitrogen atmospheric deposition"**
by H. Cathcart et al.

submitted for publication in *EGUsphere*.

General remarks:
An interesting paper describing the computation of critical loads of S and N for the whole of Canada, which is done here for the first time (earlier studies dealt with subregions only). The paper is well written, with some (minor) issues to be clarified, specially about Cl-deposition (see below). Thus, I suggest that the paper should be published with **minor revisions**. The corrections/amendments listed below should be addressed before re-submission.

Detailed remarks:
*Note: 'X' → 'Y' means: replace 'X' by 'Y' (in the text).*

**Abstract:**
L[ine] 20: "Soil critical loads of nutrient nitrogen … model.": That's already said in the first sentence!

**1 Introduction:**
L28: "… and acidic deposition …": a bit superfluous, since it is S and N deposition that forms acidic deposition … Drop or reformulate!
L46: Delete 'acidic' – this also holds for nutrient N CLs.
L57: Insert 'soil chemistry' after 'steady-state'.
L61: 'disharmony': isn't there a better word? (incompatibility?)
L72: 'effect' → 'effects'.
L73: 'estimated ' → 'assessed'.
L75: Insert '(i.e. S+N)' after 'acidic' (?).
L78: 'criterion' → 'criteria'.

**2 Methods:**
L113: Delete 'and defined as'.
L116: '$BC_{dep}$': In the text all mathematical variables are in upright font, whereas in the equations they are in italics! I suggest to make it in italics everywhere (?)
L138: The parentheses could be left away … [also in some other equations]
L152: What is 'WGS84'?
Table 1: (a) Reference(s) for 'Biomass' is/are missing;  (b) Remove parentheses around 'CEC, 2018';  same for 'Galmarini …' and 'Moran …'.
L161: What does 'Generalised' mean in this context?
L171: What do you mean by '$\times$ 5g kg$^{-1}$'? Clarify!
L182: '$Bc_{dep}$': In eq.1 it's $BC_{dep}$! But since you are never mentioning again $Cl_{dep}$ (from eq.1), I presume that you have assumed that $Na_{dep} = Cl_{dep}$, and thus $BC_{dep} - Cl_{dep}$ in eq.1 becomes $Bc_{dep}$!? **This has to be corrected/explained!**
L250: Delete 'scenarios'.
Table 2: Insert '(mol/mol)' after 'Bc/Alcrit' in header.
L263:'… all other regions were set to 0' → '… in all other regions it was set to 0'.
L278: 'against' → 'using'.
L290: Explain 'CPCAD'.

**3 Results:**

L309: 'estimate' → 'estimated'.
L326: 'under 5% and 20% growth reduction scenarios' → 'for 5% and 20% growth reductions'.
L329: 'coverage' → 'averages' (?) [otherwise, what do you mean?]
L335: $BC_{dep}$: Why suddenly upper-case 'C'?
L338: 'loss of definition': What do you mean? [is this a technical term well known?]
L338: The value after smoothing (68) is higher than the pre-smoothing value (52)?
L372: Delete '(Figure 12B)' – it shows nothing about acidity CLs.
L386/7: A bit strange text; maybe write 'Modelled annual average (2014-2016) total deposition of sulphur ($S_{dep}$, panel A) and nitrogen ($N_{dep}$, panel B). Maps were …'
L389: Insert 'average' before '2014-2016'. (?)
L394: '… result of higher CL…' → '…result of lower CL…'. !!
L403: 'plus' → 'and'. (also L409)
L407: 'including' → 'using'.
L425: 'reducing' → 'increasing'! [adding Bcdep allows more N-uptake, leading to higher CLs, and thus lower/less exceedances (5% instead of 9%)]

**4 Discussion:**
L438: 'of' → 'in' (?)
L454: 'Mcnulty' → 'McNulty'.
L458: Delete 'map', twice. (?)
L472: '7.26 ×' → '7.26 times'.
L502: 'region' → 'ecosystems'.
L522: 'Pine' → 'pine'.
L524: 'ha-1' → '$ha^{-1}$'; also for 'yr-1')

**5 Conclusions:**
L540: Delete 'to several parameters'.
L545: insert 'average' after 2014-2016.
L553: 'they provide' → 'as providing'.

**References:**
General: The references are not given in a consistent style: In some the title of the paper/report is with upper-case first letters, in some not. In some the DOI is given, in others not. In some, especially for reports and book chapters, the place and/or publisher is missing.

Specific:
L674: 'Mcnulty' → 'McNulty'.
L700/1: Is this an article? In which Journal? Or a report?
L707: Spell out "Nord. Counc Minist Cph,Den."
L763: 'AL' → 'Al'.
L773: 'de Vries' → 'De Vries'; and move up to the other 'De Vries …'!
L775: Why "McDONALD"?

Note: These are only those 'flaws' seen when glancing thru the Refs. I guess that there might be more … In any case: Please check and harmonise the References.

---

## Author Comment (AC1)

**Response to Reviewer #1's Comments**

**(egusphere-2024-2371)**

The manuscript "Estimates of critical loads and exceedances of acidity and nutrient nitrogen for mineral soils in Canada for 2014–2016 average annual sulphur and nitrogen atmospheric deposition" (egusphere-2024-2371) used national level input data and steady state mass balance model to determine a Canada wide critical load of acidity and nutrient nitrogen and determined exceedances. Although regional CL values had been developed this is the first CL values with currently available data calculated at such a scale. The values determined should facilitate environmental managers with sufficient information to further explore areas at risk. The authors have used a well-tested and recognized methodology and currently available to develop critical loads. Where used the assumptions have been declared for the most part. The data and method clearly explained but could benefit from additional information and clarifications in some areas as indicated in the detailed comments below. The manuscript is well written, equations are appropriately represented and for the most part the figures were informative and supported the results and discussion. All this being said, given that CLnutN exceedances are widespread and most noteworthy, however the authors have limited discussion of this component of the work.

Response: We thank the reviewer for their thoughtful suggestions to expand and round out the manuscript sections on methods, results and discussion. We agree that the $CL_{nut}N$ exceedance results would benefit from added detail, particularly considering their relevance to current research focusing on the eutrophication and biodiversity effects of nitrogen pollution. Suggestions on the last point are provided in the detailed comments, and by adopting those the discussion of $CL_{nut}N$ exceedances at the end of Section 4 has been expanded as follows (new text in bold):

"Widespread *$CL_{nut}N$* exceedance (found in the majority of the PA and OECM sites assessed) suggests that nutrient N may present a risk to biodiversity at many sites under protective measures. While some empirical studies of nutrient N have been done in Canada, a large knowledge gap exists for many Canadian ecosystems regarding the effect of nutrient nitrogen and their critical loads. Some work has developed on Jack pine and northern ecosystems; Vandinthner suggested that across Jack pine-dominant forests surrounding the AOSR, the biodiversity-based empirical critical load of nutrient N was 5.6 kg ha$^{-1}$ yr$^{-1}$ (400 eq ha$^{-1}$ yr$^{-1}$; Vandinther and Aherne, 2023a) which is above the maximum *$CL_{nut}N$* calculated in this study within 200 km of the AOSR (216 eq ha$^{-1}$ yr$^{-1}$). Further, in low deposition 'background' regions a biodiversity-based empirical critical load of 1.4 – 3.15 kg ha$^{-1}$ yr$^{-1}$ (100 – 225 eq ha$^{-1}$ yr$^{-1}$) was found to protect lichen communities and other N-sensitive species in Jack pine forests across Northwestern Canada (Vandinther and Aherne, 2023b); these are again higher compared to mean values in this study (e.g. for the Boreal Plain, 76 eq ha$^{-1}$ yr$^{-1}$). Empirical critical loads developed for ecoregions in Northern Saskatchewan (Murray et al., 2017) fall into a range of 88 – 123 eq ha$^{-1}$ yr$^{-1}$, again higher than values suggested by this study (e.g. 62 eq ha$^{-1}$ yr$^{-1}$ in Saskatchewan). **In comparison to these empirical values, *$CL_{nut}N$* values in the current work are lower by a factor of 2. If *$CL_{nut}N$* is doubled, only 10% of the soils assessed are in exceedance (versus 31% of soils). This reduction in the areal exceedance would in turn reduce the number of PA and OECM sites in exceedance.** While the spatial pattern of *$CL_{nut}N$* exceedances does not generally follow exceedances of critical loads of acidity, some areas

(including PA and OECM sites) in central Canada were estimated to be in exceedance of both critical loads of acidity and nutrient N, suggesting that this region may be of particular concern. **Given the largest areal exceedance is of $CL_{nut}N$, observational studies with the view of expanding Canadian ecosystem empirical critical loads would help determine how, and by how much, Canadian ecosystems are affected by $N_{dep}$ and how well these observations align with $CL_{nut}N$ in the current work. Additionally, vegetation community changepoint modelling such with the TITAN model (Baker, 2010) could help bring understanding to how Canadian ecosystems might experience elevated $N_{dep}$ with regard to changes to biodiversity."**

**Baker, M. E. and King, R. S.: A new method for detecting and interpreting biodiversity and ecological community thresholds, Methods Ecol. Evol., 1, 25–37, https://doi.org/10.1111/j.2041-210X.2009.00007.x, 2010.**

**Detailed comments:**

Lines 91-92: On average what is the soil depth for areas classified as barren where soil depth is indicated? As the purpose is to develop CLs to protect the ecosystem, what is the advantage of including these segments of the landscape?

Response: The Arctic is a region of interest but is challenging to map as much of this area has shallow soils. Much of Nunavut (e.g., Baffin Island and other Arctic islands) are classified as "barren lands" by the CEC land cover map, which is defined as "Areas characterized by bare rock, gravel, sand, silt, clay, or other earthen material, with little or no "green" vegetation present regardless of its inherent ability to support life. Generally, vegetation accounts for less than 10 percent of total cover". Generally, where mineral soil is indicated by the soil maps, soil depths are <10 cm (after removal of OM, CF, etc. from the soil profile). A notable reason for interest in critical loads in this region is the possibility of the Northwest Passage opening for shipping as climate change shrinks sea ice cover, resulting in increasing emissions of S and N from shipping vessels along this route. The benefit of the doubt has therefore been awarded to the soil depth layer where it does not indicate bare rock to provide an estimate for these ecosystems, which may be vulnerable to increased S and N deposition and may be particularly sensitive due to shallow soils. The text has been updated at line 92 as follows to briefly explain the advantage of keeping these areas:

"Areas considered "barren" by land classification were not excluded when mineral soil depth was indicated in the interest of including as much of the Arctic region as possible; as the Arctic may be greening under global climate change (Myers-Smith et al., 2020) and northern shipping routes become viable, the question of ecosystem health in this region becomes more material."

Myers-Smith, I.H., Kerby, J.T., Phoenix, G.K. et al: Complexity revealed in the greening of the Arctic. Nat. Clim. Chang. 10, 106–117, https://doi.org/10.1038/s41558-019-0688-1, 2020.

Lines 92-94: What % of the land was classified as peat and wetland. Where data are available has the 30% organic matter == peat and wetland soil classification been tested?

Response: According to the land cover classification map, only 3.7% of Canada was classified as wetland. This is an underestimation in the satellite classification, as it is generally estimated that 13% of Canada is covered by some kind of wetland (National Wetlands Working Group, 1997;

ECCC, 2019). Resolution may contribute to this underestimate in the satellite imagery (250 m). The 30% organic matter was not intended to equal the wetland classification, but rather to supplement it; it is likely that many soils classified as organic under the Canadian Soil Information System (>30% OM) are not identified under satellite imagery as wetland. Comprehensive wetland mapping across Canada is complex due to computational hindrances, scale, remoteness, and resolution of data, although some recent papers have explored the efficacy of using machine learning with satellite maps to overcome these challenges. The sentence at line 92 was modified to read:

"Since peat and wetland soil classification is difficult at a Canada-wide scale (i.e., data at the required scale are presently unavailable), and given that the satellite land cover map underestimates wetland cover (3.7% versus an expected 13% as given in (National Wetlands Working Group, 1997)), organic soils with 30% or more organic matter content were filtered out to close this gap."

National Wetlands Working Group: The Canadian Wetland Classification System, 2nd ed, edited by: Warner, B.G., Rubec, C.D.A., National Wetlands Working Group, Wetlands Research Branch, University of Waterloo, Waterloo, ON, Canada, 1997.

94-96: The authors should include the average % of mineral soil for the Hudson Plain ecozone.

Response: We have modified the sentence to read: "The Hudson Plain ecozone, which contains the world's largest contiguous wetland and is 80% wetland by cover (ECCC, 2016), was also broadly excluded from the study because of low mineral soil presence."

ECCC: Canadian Environmental Sustainability Indicators: Extent of Canada's Wetlands, ISBN: 978-0-660-05390-5, 2016.

Figure 1: The authors should clarify the need for the classification in Figure 1a. Figure 1b the resolution does not permit the reader to see all the classifications in the ledged on the map. These areas can be grouped to simplify the legend and make the map useful for discussing the results.

Response: We clarify that the purpose of the ecozones in Figure 1a with the modified text in lines 85-86: "Canada is home to a variety of climates, soils, vegetation, and geological structures that are often grouped into distinct ecozones which are often used to generalise critical loads across similar ecosystems (Figure 1A)"

This is a good suggestion for Fig. 1b, and the 15 land cover classes have now been reduced to nine classes (needleleaf forest, deciduous and mixed forest, sub-polar, shrubland, grassland, wetland, barren, water, snow and ice) to improve visibility in the figure. Urban has been removed from the image since no urban areas outside the agricultural ecumene are visible, and croplands are covered under the agricultural ecumene.

Figure 2: The authors should indicate the relevance of the four different regions.

Response: An explanatory sentence was added to the end of the paragraph text: "Exceedance calculations are divided into four regions to determine the shortest path to the critical load line along the function".

Lines 124-126:  What value(s) of gibbsite equilibrium constant is used?

Response: The $K_{gibb}$ values are given in section 2.7 of the manuscript, but a reference to the section was added here to direct the reader.

Lines 132-133:  Significant figures when converting units.  The authors should also explain why they selected the upper range value from Rosen et al (1992).

Response: Significant figures have been corrected to 35.7 eq ha$^{-1}$ yr$^{-1}$ (0.5 kg N ha$^{-1}$ yr$^{-1}$).  The text was modified to explain why 0.5 kg N ha$^{-1}$ yr$^{-1}$ was chosen:

A value of 35.7 eq ha$^{-1}$ yr$^{-1}$ (0.5 kg N ha$^{-1}$ yr$^{-1}$) was used, based on estimates of annual $N_i$ since the last glaciation by Rosen et al. (1992) and Johnson and Turner (2014) who recommended a range of 0.2 – 0.5 kg N ha$^{-1}$ yr$^{-1}$ and 0.5 – 1 kg N ha$^{-1}$ yr$^{-1}$ respectively; the midpoint (0.5 kg N ha$^{-1}$ yr$^{-1}$) was taken as a compromise.

Johnson, Dale W., and John Turner: Nitrogen budgets of forest ecosystems: a review, Forest Ecology and Management 318,370-379, https://doi.org/10.1016/j.foreco.2013.08.028, 2014.

Lines 133-134: The authors should include what value was used for soil denitrification factor and how was the value obtained.

Response: The fde and how it was obtained is discussed later in section 2.10, but a reference to the section was added to direct the reader.

Lines 150 -151:  The authors should site R version, and any libraries used as recommended by the software(s) developers.

Response: The R version (4.1.0) was added as well as the reference for the Terra (Hijmans,  2023) package.

Hijmans, R.J.: terra: Spatial Data Analysis.  R package version 1.7-23, https://CRAN.R-project.org/package=terra,2023.

R Core Team: R: A Language and Environment for Statistical Computing, version 4.1.0, R Foundation for Statistical Computing, Vienna, Austria, 2021.

Table 1:  The authors should indicate their use of air temperature to approximate soil temperature when determining base cation weathering.  Likely a good estimate given long term annual average data is used.

Response: The use of air temperature to approximate soil temperature has been added to Section 2.4: "Note that average annual air temperature was used to approximate annual average soil temperature in absence of a Canada-wide soil temperature map."

Lines 210-220: Would not a large amount of BC dep result in adding excess nutrient to the system?  The authors should mention any other consequences of incorporating anthropogenic BC dep.  This may proof important especially in areas that are exceeding nutrient N CL.

Response:

We acknowledge that large inputs of Bc dep may affect ecosystems, however N is usually the limiting nutrient for vegetation. But shifts in growing conditions could occur, so we have added the following text at line 2010:

"Pollutant $Bc_{dep}$ from industrial sources can cause shifts in soil pH, plant community and biodiversity, as well as direct damage to vegetation by dust (e.g. Mandre et al., 2008; Paal et al., 2013)."

Mandre, M., Kask, R., Pikk, J., and Ots, K.: Assessment of growth and stemwood quality of Scots pine on territory influenced by alkaline industrial dust, Environ. Monit. Assess., 138, 51–63, https://doi.org/10.1007/s10661-007-9790-3, 2008.

Paal, J., Degtjarenko, P., Suija, A., and Liira, J.: Vegetation responses to long-term alkaline cement dust pollution in inus sylvestris-dominated boreal forests – niche breadth along the soil pH gradient, Appl. Veg. Sci., 16, 248–259, https://doi.org/10.1111/j.1654-109X.2012.01224.x, 2013.

Lines 214-219: It is not clear the purpose of smoothing BC dep. It does not represent actual, nor does it represent non-anthropogenic BC dep.

Response: The purpose of smoothing base cation deposition was to remove the anthropogenic influence of a number of point sources (e.g., surface mines) which, when unsmoothed, resulted in very high local critical loads around the point source. In most cases it was successful in smoothing those (relatively) smaller point sources; however, the extent of anthropogenic BCdep produced by the AOSR was not mitigated very much, which may give the reader the impression that the exercise was not worthwhile. We have updated the text to elaborate on the motivations for this at line 207:

"The modelled $Bc_{dep}$ and station observations include anthropogenic input, but the $Bc_{dep}$ input to the SMB model is meant to reflect long-term non-anthropogenic sources of base cations. However, large point sources of $Bc_{dep}$ **(such as surface mines)** are a feature of some Canadian regions, and their impact should not be overlooked in critical load assessments. To demonstrate the relative impact of anthropogenic sources on Canadian critical loads estimates **and to mitigate the impact anthropogenic local Bc_dep inputs have in remote regions**, two scenarios were assessed, one including anthropogenic $Bc_{dep}$ and another that attempted to smooth out anthropogenic "hot spots"."

Lines 217-219: The work could not consider forest fire impact, and the authors have mentioned the impact of this omission on BCdep. The authors should also indicate how not considering forest fires could affect other terms in their work.

Response: We have added some text at the end of line 219 to note the important impact of fires on nitrogen removal:

"The loss of nitrogen due to forest fires from forest biomass and organic soil content is also significant (and is not reflected in $N_{up}$, which only deals with loss from harvesting)."

Lines 242-243: Was the most sensitive species given priority in a grid cell irrespective of coverage, what were the range of coverage?

Response: The grid cell was assigned a tree species if forest cover was above 25% and if the most sensitive species was present above 5% species composition.  The text at lines 242-243 has been revised as follows:

"If forest was present above 25% coverage, values were sorted by the most sensitive tree species (those with the lowest $Bc/Al_{crit}$) above 5% species composition and given priority for the 250 m grid-cell value."

Table 3:  It is not clear how V2.2 and V3.2 adds to the material presented in the section or elsewhere.

Response: There are two major versions of the Canadian Soil Information System, and they are not compatible in their geographic extent/alignment and have some drainage classification differences.  It was therefore necessary to make a crosswalk across the versions and different drainage classifications, and this was simply to document how the classifications were aligned to the table in CLRTAP (2015).  Section 2.10 has been updated as follows to clarify the need for this crosswalk:

"The soil denitrification fraction ($f_{de}$) is generally related to soil drainage (CLRTAP, 2015); classes ranging from excessive to very poor drainage were assigned using the Canada-wide Canadian Soil Information Service (CanSIS) databases v2.2 (CLBBR, 1996) and v.3.2 (SLCWG, 2010). **Because the databases are not compatible in their geographic extent and alignment,** boundary and classification priority was given to the most recent database version before rasterization. **Differences in classifications and their alignment to the soil drainage classes from CLRTAP (2015) are shown in Table 3."**

 Lines 315-317: The authors should explain/define temperature correction in the method section.

Response: As suggested, the temperature correction has been explained by an update to the text at line 164:

"A temperature correction was applied to the $BC_{we}$ as the speed of chemical weathering can be affected by temperature.  Weathering is modified by ambient temperature T...".

Table 4:  The results in the text is presented by administrative boundary and a few times by ecozones making reference to the table difficult.  Because much of the discussion is by administrative boundary much of the results (Ecozones) presented are not discussed.  What would be the takeaway for an international audience?

Response: We focused heavily on the administrative boundaries to be able to compare to provincial estimates, but we concede this may not be intuitive for international readers.  Some text has been added to the results to highlight the critical loads of notable ecozones:

At line 373: "From an ecozone perspective the Mixedwood Plain ecozone had the highest $CL_{max}S$ at 1586 eq ha$^{-1}$ yr$^{-1}$ followed by the Prairies at 1078 eq ha$^{-1}$ yr$^{-1}$. The most sensitive ecozones outside the Arctic ecozones (which were below 100 eq ha$^{-1}$ yr$^{-1}$) were the Boreal Cordillera and the Taiga ecozones (Table 4).

The coastal ecozones had the highest $CL_{nut}N$, with the Pacific and Atlantic Maritime zones having 513 and 235 eq ha$^{-1}$ yr$^{-1}$ respectively. The Prairie ecozone had the lowest $CL_{nut}N$, lower than some of the Arctic ecozones, at 63 eq ha$^{-1}$ yr$^{-1}$."

Lines 337-339: The average base cation deposition is higher after smoothing?

Response: This is an error in the text, in which the base cation deposition was mistakenly switched in the text. The average Bcdep is now corrected to show that it is lower after smoothing.

Lines 339-340: Where is the star on the map located presumably the entire region is not a source of BC deposition?

Response: To clarify the star represents a point within the AOSR, the Figure 5 caption has been updated to read "The location of the city of Fort McMurray within the Athabasca Oil Sands Region (AOSR) is indicated by a star."

Section 3.5: Why does this section not include CLnutN results?

Response: We initially included the CLnutN results in this section and later moved them to the concluding paragraph in section 3.7 for cohesiveness with figures. The first paragraph of the text describing CLnutN in section 3.7 has been moved back to section 3.5 with the additions noted in the comment above, while keeping the description of CLnutN exceedance in 3.7. The figures have been renumbered to accommodate the shift.

Lines 368-369: Which BC deposition is used to determine CLmaxS? Earlier when discussing the BC deposition (smoothed unsmoothed), the authors should indicate which of the two data sets is the primary data set.

Response: The CLmaxS produced with the smoothed Bcdep was the one used at line 368, and this has been clarified earlier (in section 3.2) as recommended: "The smoothed Bcdep was adopted as the primary data set for presenting the critical loads."

Lines 379-382: Where are the isolated point sources and what are they due to? These seem to be in different areas of the province outside the previously mentioned AOSR.

Response: The major ones are Sudbury, Ontario and Thompson, Manitoba (nickel smelting and mining). The Thompson smelter closed in 2015 and the mine closed in 2017. The sentence at line 382 has been modified to note these examples:

"Southern Quebec and central Ontario showed higher annual average values between 50–200 eq ha$^{-1}$ yr$^{-1}$, with some point sources showing Sdep in excess of 500 eq ha$^{-1}$ yr$^{-1}$ (e.g., at nickel smelters and mining operations in Sudbury, Ontario and Thompson, Manitoba)."

Lines 382-384: The authors discuss Ndep by provincial average, but the most sticking observation is the north-south gradient. The authors should discuss what is driving the observed north-south gradient observed for much of the country.

Response: A short description of the north-south gradient was added to this section: "A north-south Ndep gradient is observable in Figure 9B, showing higher Ndep closer to agricultural sources in southern Ontario and Quebec and in the Prairies."

Lines 393-395: Would not the CL values be lower for 5% protection?

Response: This is an error in the text and has been corrected to read higher values for 5% protection.

Lines 425-426: It seems like a great many of the exceedances are less than 50 eq ha$^{-1}$ yr$^{-1}$ and probably within the uncertainty of the model. In addition to absolute exceedances the authors should comment on the various ranges.

Response: It is likely that the low (but widespread) exceedances are within the uncertainty of the model, and we have added some text to the discussion paragraph at line 519:

"However, 40% of the grid cells showing $CL_{nut}N$ exceedance were below 50 eq ha-1 yr-1 and it is likely that many of these exceedances are within the uncertainty of the model."

We have also adopted the suggestion to comment on the range and have added some detail to address this after line 517:

"Some regions adjacent to the agricultural ecumene in the Prairies, southern Ontario, Quebec and the AOSR experienced values above 300 eq ha$^{-1}$ yr$^{-1}$ up to 1053 eq ha$^{-1}$ yr-1; however, 80% of grid cells in exceedance fell below 300 eq ha$^{-1}$ yr$^{-1}$ (Figure 12B).

Lines 486-489: The choice of level of protection for a managing agency is really a policy decision and not an ethical choice.

Response: We concede and have clarified the sentence to read: "It should be noted that the level of protection is a policy decision regarding how much should be protected, rather than a sensitivity …"

Line 491: Low CLnut N is more widespread than the Arctic, the authors should indicate what is driving the low values in other areas not just the arctic.

Response: Low Q values extend from the Arctic down to central Canada; $CL_{nut}N$ seems to be driven more by Q than by vegetation cover. The sentence has been modified to clarify that Q-driven low $CL_{nut}N$ is not confined to the Arctic, but also to central Canada where Q values are below 50 mm. The paragraph at 491 has been reworked to read:

CLnutN seems to be driven primarily by $Q$ rather than by vegetation cover; low $CL_{nut}N$ was seen in regions of correspondingly low $Q$ values (e.g., >50 mm yr$^{-1}$) in much of the Arctic and central

Canada. In contrast, areas with high $Q$ were found to result in high $CL_{nut}N$; as previously suggested by Reinds et al. (2015), a critical flux rather than concentration may provide more reliable critical loads in regions with elevated precipitation such as the Pacific Maritime ecozone in British Columbia.

Lines 522:534:  The comparison works presented seem to indicate the current work is under estimating CLnutN by a factor of 2, relative to the presented comparison work.  What is the % of area with deposition greater than 2x CLnutN? See also comment on discussing various ranges of CLnutN exceedances.

Response:  This is an interesting consideration. 10% of the area has an $N_{dep}$ greater than 2 x CLnutN, and most of this area is adjacent to the agricultural ecumene (and corresponds to the areas of >300 eq ha$^{-1}$ yr$^{-1}$ exceedance).  The following text has been added to the discussion (around line 531):

**In comparison to these empirical values, $CL_{nut}N$ values in the current work are lower by a factor of 2.  If $CL_{nut}N$ is doubled, only 10% of the soils assessed are in exceedance (versus 31% of soils).  This reduction in the areal exceedance would in turn reduce the number of PA and OECM sites in exceedance.**

Line 531-534:  it looks like exceedance is largely driven by Ndep as CLnutN is very low in much of the area under study.  Given the largest exceendance is of CLnutN what additional work or data do the authors recommend?

Response: We would like to suggest the following new text, added after line 534:

"Given the largest areal exceedance is of $CL_{nut}N$, observational studies with the view of expanding Canadian ecosystem empirical critical loads would help determine how, and by how much, Canadian ecosystems are affected by $N_{dep}$ and how well these observations align with $CL_{nut}N$ as produced in this manuscript.  Additionally, vegetation community changepoint modelling such with the TITAN model (Baker, 2010) could help bring understanding to how Canadian ecosystems might experience elevated $N_{dep}$ with regard to changes in biodiversity."

Baker, M. E., and R. S. King: A new method for detecting and interpreting biodiversity and ecological community thresholds, Methods Ecol. Evol. 1, 25–37, 2010.

---

## Author Comment (AC2)

**Response to Reviewer #2's Comments**

**(egusphere-2024-2371)**

**General remarks:**

An interesting paper describing the computation of critical loads of S and N for the whole of Canada, which is done here for the first time (earlier studies dealt with subregions only). The paper is well written, with some (minor) issues to be clarified, specially about Cl-deposition (see below). Thus, I suggest that the paper should be published with minor revisions. The corrections/amendments listed below should be addressed before re-submission.

Response: We would like to thank this reviewer for a very thorough parsing of the text and the identification of many minor issues that on correction will greatly improve readability and understanding.

The Cl deposition with regards to the issue of BCdep/Bcdep has been clarified in the methods section with the following explanation: The assumption is that Nadep = Cldep, in the absence of a Cldep map. We acknowledge this needs clarification and have added the following sentence to section 2.5 (Base cation deposition):

"In the absence of a modelled $Cl_{dep}$ map, and since the air quality model estimates non-marine $BC_{dep}$, $Cl_{dep}$ was assumed to equal sodium deposition; $BC_{dep}$ is therefore referred to as $Bc_{dep}$)."

**Detailed remarks:**
Note: 'X'→ 'Y' means: replace 'X' by 'Y' (in the text).

Abstract:
L[ine] 20: "Soil critical loads of nutrient nitrogen ... model.": That's already said in the first sentence!
Response: It is repetitive, the sentence in line 20 has been removed.

1 Introduction:
L28: "... and acidic deposition ...": a bit superfluous, since it is S and N deposition that forms acidic deposition ... Drop or reformulate!
Response: The sentence has been reformulated to "...reductions in sulphur (S)) and nitrogen (N) deposition..."

L46: Delete 'acidic' – this also holds for nutrient N CLs.
Response: "Acidic" has been deleted from the sentence.

L57: Insert 'soil chemistry' after 'steady-state'.
Response: "Soil chemistry" has been added to the sentence.

L61: 'disharmony': isn't there a better word? (incompatibility?)
Response: Incompatibility is a good suggestion, it has replaced "disharmony".

L72: 'effect' → 'effects'.
Response: "Effect" has been updated to "effects".

L73: 'estimated ' → 'assessed'.
Response: "Estimated" has been updated to "assessed".

L75: Insert '(i.e. S+N)' after 'acidic' (?).
Response: S+N has been inserted after "acidic".

L78: 'criterion' → 'criteria'.
Response: "criteria" has been updated to "criterion".

2 Methods:
L113: Delete 'and defined as'.
Response: "and defined as" was removed.

L116: 'BCdep': In the text all mathematical variables are in upright font, whereas in the equations they are in italics! I suggest to make it in italics everywhere (?)
Response: The manuscript has been updated to make the mathematical variables that appear within the text in italics.

L138: The parentheses could be left away ... [also in some other equations]
Response: As suggested, the parentheses were removed from eq. 4 and 5 but retained in eq 2 ($ANC_{le,crit}$) for readability (the exponents otherwise look strange).

L152: What is 'WGS84'?
Response: The sentence has been updated to clarify WGS84 is the World Geodesic System 1984 projection system of the map.

Table 1: (a) Reference(s) for 'Biomass' is/are missing; (b) Remove parentheses around 'CEC, 2018'; same for 'Galmarini ...' and 'Moran ...'.
Response: The biomass references have been added and the parentheses removed.

L161: What does 'Generalised' mean in this context?
Response: This was poorly worded and "generalised" has been removed from the sentence.

L171: What do you mean by '× 5g kg-1'? Clarify!
The organic carbon map sourced from Hengl and Wheeler is provided in units x5g kg-1. This is perhaps too much unnecessary detail and the conversion is provided in the data notes provided by Hengl and Wheeler, so we have removed the reference to x5g kg-1 to avoid confusion.

L182: 'Bcdep': In eq.1 it's BCdep! But since you are never mentioning again Cldep (from eq.1), I presume that you have assumed that Nadep = Cldep, and thus BCdep – Cldep in eq.1 becomes Bcdep!? This has to be corrected/explained!
Response: Yes, the assumption is that Nadep = Cldep, in the absence of a Cldep map. We acknowledge this needs clarification and have added the following sentence to section 2.5 (Base cation deposition) at line 194:

"In the absence of a modelled $Cl_{dep}$ map, and since the air quality model estimates non-marine $BC_{dep}$, $Cl_{dep}$ was assumed to equal sodium deposition; $BC_{dep}$ is therefore referred to as $Bc_{dep}$)."

L250: Delete 'scenarios'.
Response: "Scenarios" was removed from the caption.

Table 2: Insert '(mol/mol)' after 'Bc/Alcrit' in header.
Response: The table has been updated to add mol/mol.

L263:'... all other regions were set to 0' → '... in all other regions it was set to 0'.
Response: The text was changed to "in all other regions it was..."

L278: 'against' → 'using'.
Response: "against" was replaced by "using".

L290: Explain 'CPCAD'.
Response: CPCAD was defined in the introduction; however, it has been re-defined here for clarity (Canadian Protected and Conserved Areas Database).

3 Results:

L309: 'estimate' → 'estimated'.
Response: "estimate" was changed to "estimated".

L326: 'under 5% and 20% growth reduction scenarios' → 'for 5% and 20% growth reductions'.
Response:  This suggestion was implemented.

L329: 'coverage' → 'averages' (?) [otherwise, what do you mean?]
Response: Yes, thank you, this has been changed to "averages".

L335: BCdep: Why suddenly upper-case 'C'?
Response: This is a typo, it has been corrected to lower-case Bcdep.

L338: 'loss of definition': What do you mean? [is this a technical term well known?]
Response: This sentence has been reworked to read "at the cost of lowering average Bcdep".

L338: The value after smoothing (68) is higher than the pre-smoothing value (52)?
Response: This was an error in the text, and has been corrected to show the after smoothing average is lower than the pre-smoothing average.

L372: Delete '(Figure 12B)' – it shows nothing about acidity CLs.
Response: Figure 12B was removed from this line.

L386/7: A bit strange text; maybe write 'Modelled annual average (2014-2016) total deposition of sulphur (Sdep, panel A) and nitrogen (Ndep, panel B). Maps were ...'
Response: This is a good reworking, it has been adopted.

L389: Insert 'average' before '2014-2016'. (?)
Response: added "average" before "2014-2016".

L394: '... result of higher CL...' → '...result of lower CL...'. !!
Response: Thank you for catching this terrible reversal, it has been changed to "lower"

L403: 'plus' → 'and'. (also L409)
Response: Good distinction, "plus" was changed to "and" in both these lines.

L407: 'including' → 'using'.
Response: "Including" was changed to "using".

L425: 'reducing' → 'increasing'! [adding Bcdep allows more N-uptake, leading to higher CLs, and thus lower/less exceedances (5% instead of 9%)]
Response: Thank you for catching another reversal, it has been changed to "increasing".

4 Discussion:

L438: 'of' → 'in' (?)
Response: "In" sounds much nicer, "of" has been changed to "in".

L454: 'Mcnulty' → 'McNulty'.
Response: McNulty has been properly formatted here and in the references.

L458: Delete 'map', twice. (?)
Response: The second "map" in the sentence was removed.

L472: '7.26 ×' → '7.26 times'.
Response: The x was reformatted to "times".

L502: 'region' → 'ecosystems'.
Response: "Ecosystems" replaced "the region".

L522: 'Pine' → 'pine'.
Response: The capital was corrected.

L524: 'ha-1' → 'ha$^{-1}$'; also for 'yr-1')
Response: The superscripts have been corrected.

5 Conclusions:

L540: Delete 'to several parameters'.
Response: "to several parameters" was removed.

L545: insert 'average' after 2014-2016.
Response: "average" was inserted after 2014-2016.

L553: 'they provide' → 'as providing'.

Response: "they provide" was changed to "as providing".

References:

General: The references are not given in a consistent style: In some the title of the paper/report is with upper-case first letters, in some not. In some the DOI is given, in others not. In some, especially for reports and book chapters, the place and/or publisher is missing.

Specific:
L674: 'Mcnulty' → 'McNulty'.
L700/1: Is this an article? In which Journal? Or a report?
L707: Spell out "Nord. Counc Minist Cph,Den."
L763: 'AL' → 'Al'.
L773: 'de Vries' → 'De Vries'; and move up to the other 'De Vries …'!
L775: Why "McDONALD"?
Note: These are only those 'flaws' seen when glancing thru the Refs. I guess that there might be more … In any case: Please check and harmonise the References

Response: Yes, it seems that a number of the references that were generated using a referencing software package were poorly formatted.  In addition to fixing the above noted errors, the references have been checked and manually re-formatted as necessary.